# PGRAD : LEARNING PRINCIPAL GRADIENTS FOR DOMAIN GENERALIZATION

**Zhe Wang, Jake Grigsby, Yanjun Qi**
Department of Computer Science
University of Virginia
Charlottesville, VA 22903, USA
{zw6sg, jcg6dn, yq2h}@virginia.edu

## ABSTRACT

Machine learning models fail to perform when facing out-of-distribution (OOD) domains, a challenging task known as domain generalization (DG). In this work, we develop a novel DG training strategy, we call PGrad, to learn a robust gradient direction, improving models' generalization ability on unseen domains. The proposed gradient aggregates the principal directions of a sampled roll-out optimization trajectory that measures the training dynamics across all training domains. PGrad's gradient design forces the DG training to ignore domain-dependent noise signals and updates all training domains with a robust direction covering main components of parameter dynamics. We further improve PGrad via bijection-based computational refinement and directional plus length-based calibrations. Our theoretical proof connects PGrad to the spectral analysis of Hessian in training neural networks. Experiments on DomainBed and WILDs benchmarks demonstrate that our approach effectively enables robust DG optimization and leads to smoothly decreased loss curves. Empirically, PGrad achieves competitive results across seven datasets, demonstrating its efficacy across both synthetic and real-world distributional shifts.

## 1 INTRODUCTION

Deep neural networks have shown remarkable generalization ability on test data following the same distribution as their training data. Yet, high-capacity models are incentivized to exploit any correlation in the training data that will lead to more accurate predictions. As a result, these models risk becoming overly reliant on "domain-specific" correlations that may harm model performance on test cases from out-of-distribution (OOD). A typical example is a camel-and-cows classification task (Beery et al., 2018; Shi et al., 2021), where camel pictures in training are almost always shown in a desert environment while cow pictures mostly have green grassland backgrounds. A typical machine learning model trained on this dataset will perform worse than random guessing on those test pictures with cows in deserts or camels in pastures. The network has learned to use the background texture as one deciding factor when we want it to learn to recognize animal shapes. Unfortunately, the model overfits to specific traps that are highly predictive of some training domains but fail on OOD target domains. Recent domain generalization (DG) research efforts deal with such a challenge. They are concerned with how to learn a machine learning model that can generalize to an unseen test distribution when given multiple different but related training domains. [1]

Recent literature covers a wide spectrum of DG methods, including invariant representation learning, meta-learning, data augmentation, ensemble learning, and gradient manipulation (more details in Section 2.4). Despite the large body of recent DG literature, the authors of (Gulrajani & Lopez-Paz, 2021) showed that empirical risk minimization (ERM) provides a competitive baseline on many real-world DG benchmarks. ERM does not explicitly address distributional shifts during training. Instead, ERM calculates the gradient from each training domain and updates a model with the average gradient. However, one caveat of ERM is its average gradient-based model update will preserve domain-specific noise during optimization. This observation motivates the core design of our method.

---

[1]In the rest of this paper, we use the terms "domain" and "distribution" interchangeably.

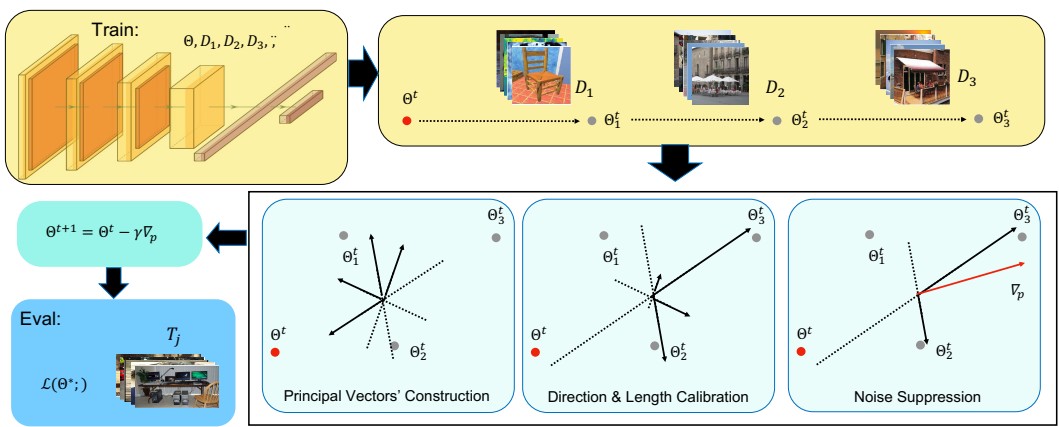

Figure 1: Overview of our PGrad training strategy. With a current parameter $\Theta^t$, we first obtain a rollout trajectory $\Theta^t \to \Theta_1^t \to \Theta_2^t \to \Theta_3^t$ by sequentially optimizing across all training domains $\mathcal{D}_{tr} = \{D_i\}_{i=1}^3$. Then PGrad updates $\Theta^t$ by extracting the principal gradient direction $\nabla_p$ of the trajectory. A target model's generalization is evaluated on unseen (OOD) test domains $T_j$.

We propose a novel training strategy that learns a robust gradient direction for DG optimization, and we call it PGrad . PGrad samples an optimization trajectory in high dimensional parameter space by updating the current model sequentially across training domains. It then constructs a local coordinate system to explain the parameter variations in the trajectory. Via singular value decomposition (SVD), we derive an aggregated vector that covers the main components of parameter dynamics and use it as a new gradient direction to update the target model. This novel vector - that we name the "principal gradient" - reduces domain-specific noise in the DG model update and prevents the model from overfitting to particular training domains. To decrease the computational complexity of SVD, we construct a bijection between the parameter space and a low-dimensional space through transpose mapping. Hence, the computational complexity of the PGrad relates to the number of sampled training domains and does not depend on the size of our model parameters.

This paper makes the following contributions: (1) PGrad places no explicit assumption on either the joint or the marginal distributions. (2) PGrad is model-agnostic and is scalable to various model architecture since its computational cost only relates to the number of training domains. (3) We theoretically show the connection between PGrad and Hessian approximation, and also prove that PGrad benefits the training efficiency via learning a gradient in a smaller subspace constructed from learning trajectory. (4) Our empirical results demonstrate the competitive performance of PGrad across seven datasets covering both synthetic and real-world distributional shifts.

## 2 METHOD

Domain generalization (Wang et al., 2021; Zhou et al., 2021) assumes no access to instances from future unseen domains. In domain generalization, we are given a set of training domains $\mathcal{D}_{tr} = \{D_i\}_{i=1}^n$ and test domains $\mathcal{T}_{te} = \{T_j\}_{j=1}^m$. Each domain $D_i$ (or $T_j$) is associated with a joint distribution $\mathcal{P}_{\mathcal{X} \times \mathcal{Y}}^{D_i}$ (or $\mathcal{P}_{\mathcal{X} \times \mathcal{Y}}^{T_j}$), where $\mathcal{X}$ represents the input space and $\mathcal{Y}$ is the output space. Moreover, each training domain $D_i$ is characterized by a set of i.i.d samples $\{\mathbf{x}_k^i, \mathbf{y}_k^i\}$. For any two different domains sampled from either $\mathcal{D}_{tr}$ or $\mathcal{T}_{te}$, their joint distribution varies $\mathcal{P}_{\mathcal{X} \times \mathcal{Y}}^{D_i} \neq \mathcal{P}_{\mathcal{X} \times \mathcal{Y}}^{D_j}$, and most importantly, $\mathcal{P}_{\mathcal{X} \times \mathcal{Y}}^{D_i} \neq \mathcal{P}_{\mathcal{X} \times \mathcal{Y}}^{T_j}$.

We consider the prediction task from the input $\mathbf{x} \in \mathcal{X}$ to the output $\mathbf{y} \in \mathcal{Y}$. Provided with a model family whose parameter space is $\Theta \subset \mathbb{R}^d$ and the loss function $\mathcal{L} : \Theta \times (\mathcal{X} \times \mathcal{Y}) \to \mathbb{R}_+$, the goal is to find an optimal $\Theta_{te}^*$ on test domains so that:

$$\Theta_{te}^* = \arg\min_{\Theta \in \Theta} \mathbb{E}_{T_j \sim \mathcal{T}_{te}} \mathbb{E}_{(\mathbf{x},\mathbf{y}) \sim \mathcal{P}_{\mathcal{X} \times \mathcal{Y}}^{T_j}} \mathcal{L}[\Theta, (\mathbf{x}, \mathbf{y})]. \tag{1}$$

In DG setup, any prior about $\mathcal{T}_{te}$, such as inputs or outputs, are unavailable in the training phase.

Despite not considering domain discrepancies from training to testing, ERM is still a competitive method for domain generalization tasks (Gulrajani & Lopez-Paz, 2021). ERM naively groups data

from all training domains $\mathcal{D}_{tr}$ together and obtains its optimal parameter $\Theta_{tr}^*$ via the following to approximate $\Theta_{te}^*$:

$$\Theta_{tr}^* = \arg\min_{\Theta \in \mathbf{\Theta}} \mathbb{E}_{D_i \sim \mathcal{D}_{tr}} \mathbb{E}_{(\mathbf{x}, \mathbf{y}) \sim \mathcal{P}_{\mathcal{X} \times \mathcal{Y}}^{D_i}} \mathcal{L}[\Theta, (\mathbf{x}, \mathbf{y})]. \tag{2}$$

In the rest of the paper, we omit the subscript in $\Theta_{tr}$ and use $\Theta$ for simplicity (during DG training, only training domains $\mathcal{D}_{tr}$ will be available for model learning).

When optimizing with ERM on DG across multiple training domains, the update of $\Theta$ follows:

$$\Theta^{t+1} = \Theta^t - \frac{\gamma}{n} \sum_{i=1}^{n} \nabla_{\Theta^t} \mathcal{L}_{D_i}, \tag{3}$$

where $\nabla_{\Theta^t} \mathcal{L}_{D_i} = \nabla \mathbb{E}_{(\mathbf{x}, \mathbf{y}) \sim \mathcal{P}_{\mathcal{X} \times \mathcal{Y}}^{D_i}} \mathcal{L}[\Theta^t, (\mathbf{x}, \mathbf{y})]$ calculates the gradient of the loss on domain $D_i$ with respect to the current parameter $\Theta^t$ and $\gamma$ is the learning rate.

The gradient determines the learning path of a model. When using ERM in DG setting, each step of model updates uses an average gradient and may introduce and preserve domain-specific noise. For instance, if one training domain includes the trapping signals like cows always in pastures and camels always in deserts (as mentioned earlier). When investigating across multiple training domains, we envision such domain-specific noise signals will not be the main components of parameter variations across all domains. This motivates us to design PGrad as follows.

## 2.1 PGrad : Principal Gradient based Model Updates

We extend ERM with a robust gradient estimation that we call PGrad . We visualize an overview in Figure 1 to better explain how it works. Briefly speaking, given the current model parameter vector, we sample a trajectory of parameters by training sequentially on each training domain. Next, we build a local principal coordinate system based on parameters obtained from the sampled trajectory. The chosen gradient direction is then built as a linear combination of the orthogonal axes of the local principal coordinates. Our design also forces the learned gradient to filter out domain-specific noise and follows a direction that maximally benefits all training domains $\mathcal{D}_{tr}$; we refer to this extracted direction as the principal gradient. In the following, we cover details of the trajectory sampling, local coordinate construction, direction and length calibration, plus the noise suppression for our principal gradient design.

**Trajectory Sampling.** Denote the current parameter vector as $\Theta^t$. We first sample a trajectory $\boldsymbol{S}$ through the parameter space $\Theta$ by sequentially optimizing the model on each training domain:

$$\Theta_0^t = \Theta^t, \ \Theta_i^t = \Theta_{i-1}^t - \alpha \nabla_{\Theta_{i-1}^t} \mathcal{L}_{D_i}, \ i = \{1, \cdots, n\} \tag{4}$$

We refer to the process of choosing an order of training domain to optimize as *trajectory sampling*. Different ordering arrangements of training domains will generate different trajectories.

**Principal Coordinate system Construction.** Now we have a sampled trajectory $\boldsymbol{S} = \{\Theta_i^t\}_{i=0}^n \in \mathbb{R}^{(n+1) \times d}$, that were derived from $\Theta^t$. Note: the inclusion of the starting location $\Theta_0^t$ as part of the trajectory is necessary; see the proof in Appendix (A.3).

Then we construct a local principal coordinate system to explain the variations in $\boldsymbol{S}$. We are looking for orthogonal and unit axes $\boldsymbol{V} = [\boldsymbol{v}_z^\mathsf{T}]_{z=0}^n \in \mathbb{R}^{(n+1) \times d}$ to maximally capture the variations of the trajectory. Each $\boldsymbol{v}_z \in \mathbb{R}^d$ is a unit vector of size $d$, the same dimension as the parameters $\Theta^t$.

$$\max_{\boldsymbol{v}_z} \mathbf{Variance}([\boldsymbol{S}\boldsymbol{v}_z]), \ s.t. \ \boldsymbol{V}^T \boldsymbol{V} = \mathbf{I}_d. \tag{5}$$

The above objective is the classic principal component analysis formulation and can be solved with singular value decomposition (a.k.a. SVD). Eq. (5)) has the closed-form solution as follows (the revised computational complexity is $n$):

$$\lambda_z, \ \boldsymbol{v}_z = \mathbf{SVD}_z(\frac{1}{n} \hat{\boldsymbol{S}}^T \hat{\boldsymbol{S}}), \tag{6}$$

Here $\lambda_z, \boldsymbol{v}_z$ denote the $z$-th largest eigenvalue and its corresponding eigenvector. $\hat{\boldsymbol{S}}$ denotes the centered trajectory by removing the mean from $\boldsymbol{S}$. In the above Eq. (6)), the computational bottleneck lies in the SVD, whose computational complexity comes at $\mathcal{O}(d^3)$ due to $\hat{\boldsymbol{S}}^T \hat{\boldsymbol{S}} \in \mathbb{R}^{d \times d}$. $d$ denotes

the size of the parameter vector and is fairly large for most state-of-the-art (SOTA) deep learning architectures. This prohibits the computation of the eigenvalues and eigenvectors from Eq. (6) for SOTA deep learning models. Hence, we refine and construct a bijection as follows to lower the computational complexity (to $\mathcal{O}((n+1)^3)$):

$$\hat{S}\hat{S}^T \boldsymbol{e}_z = \lambda_z \boldsymbol{e}_z \quad \Longrightarrow \quad \hat{S}^T \hat{S}\hat{S}^T \boldsymbol{e}_z = \lambda_z \hat{S}^T \boldsymbol{e}_z \quad \Longrightarrow \quad \boldsymbol{v}_z = \hat{S}^T \boldsymbol{e}_z \tag{7}$$

Eq. (7) indicates that if $\lambda_z, \boldsymbol{e}_z$ are the $z$-th largest eigenvalue and corresponding eigenvector of $\hat{S}\hat{S}^T$, the $z$-th largest eigenvalue and corresponding eigenvector of $\hat{S}^T \hat{S}$ are $\lambda_z, \hat{S}^T \boldsymbol{e}_z$ (i.e., $\boldsymbol{v}_z = \hat{S}^T \boldsymbol{e}_z$). This property introduces a bijection from eigenvectors of $\hat{S}\hat{S}^T \in \mathbb{R}^{(n+1)\times(n+1)}$ to those of $\hat{S}^T \hat{S} \in \mathbb{R}^{d\times d}$. Since $n$ - the number of training domains - is much smaller than $d$, calculating eigen-decomposition of $\hat{S}\hat{S}^T \in \mathbb{R}^{(n+1)\times(n+1)}$ is therefore much cheaper.

**Directional Calibration.** With the derived orthogonal axes $\boldsymbol{V} = [\boldsymbol{v}_z^\mathsf{T}]_{z=0}^n$ from Eq. (7), now we construct a local principal coordinate system with each axis aligning with one eigenvector $\boldsymbol{v}_z$. These principal coordinate axes $\boldsymbol{V}$ are ordered based on the magnitude of the eigenvalues. This means that $\boldsymbol{v}_i$ explains more variations of the sampled trajectory $\boldsymbol{S}$ than $\boldsymbol{v}_j$ if $i < j$, and they are all unit vectors. In addition, these vectors are unoriented, which means either positive or negative multiple of an eigenvector still falls into the eigenvector set.

Now our main goal is to get a robust gradient direction by aggregating information from $\boldsymbol{V}$. First we calibrate the directions of each eigenvectors so that they point to the directions that can improve the DG prediction accuracy. Learning an ideal direction is impossible without a reference. The choice of the reference is flexible, as long as it is pointing to a direction that climbs up the loss surface. We want the reference to guide the principal gradient in the right direction for gradient descent based algorithms. For simplicity, we use the difference between the starting point $\Theta_0^t$ and the end point $\Theta_n^t$ of the trajectory $\boldsymbol{S}$ as our reference $\nabla_r = \Theta_0^t - \Theta_n^t$. So for each coordinate axis, we revise its direction so that the resulting vector $\boldsymbol{w}_z$ is positively related to the reference $\nabla_r$ in terms of the inner product:

$$\boldsymbol{w}_z = r_z \boldsymbol{v}_z, \;\; r_z = \begin{cases} 1, & \text{if } \langle \boldsymbol{v}_z, \nabla_r \rangle \geq 0, \\ -1, & \text{otherwise.} \end{cases} \tag{8}$$

**Constructing Principal Gradient.** The relative importance of each $\boldsymbol{w}_z$ is conveyed in the corresponding eigenvalue $\lambda_z$. Larger $\lambda_z$ implies higher variance when projecting the trajectory $\boldsymbol{S}$ along $\boldsymbol{w}_z$ direction. We weight each axis with their eigenvalues, and aggregate them together into a weighted sum. This gives us the principal gradient vector being calculated as follows:

$$\nabla_p = \sum_{z=0}^n \frac{\lambda_z}{||\boldsymbol{\lambda}||_2} \boldsymbol{w}_z, \;\; \boldsymbol{\lambda} = [\lambda_0, \lambda_1, \cdots, \lambda_n] \tag{9}$$

There exists other possible aggregation besides Eq. (9). For instance, another choice of weight could be $\lambda_z/||\boldsymbol{\lambda}||_1$ or simply $\lambda_z$ since the eigenvalue of a semi-positive definite matrix is non-negative. Gradient normalization has been widely recommended for improving training stability (You et al., 2017; Yu et al., 2017). Our design in Eq. (9) automatically achieves $L_2$ normalization, because:

$$||\nabla_p||_2^2 = \sum_{z=0}^n \frac{\lambda_z^2}{||\boldsymbol{\lambda}||_2^2} ||\boldsymbol{w}_z||_2^2 = 1, \tag{10}$$

**Length Calibration.** As training updates continue, a fixed length gradient operator may become too rigid, causing fluctuations in the loss. We, therefore, propose to calibrate the norm of $\nabla_p$ with a reference, for achieving adaptive length tuning. Specifically, we propose to multiply the aggregated gradient from Eq. (9) with the $L_2$ norm of $\nabla_r$:

$$\nabla_p = \sum_{z=0}^n \frac{\lambda_z ||\nabla_r||_2}{||\boldsymbol{\lambda}||_2} \boldsymbol{w}_z, \tag{11}$$

With this length calibration via $||\nabla_r||_2$, the norm of the proposed gradient is constrained by the multiplier, and is automatically tuned during the training process.

**Noise Suppression.** Most $\boldsymbol{w}_z$ axes correspond to small eigenvalues and may relate to just domain-specific noise signals. They may help the accuracy of a specific training domain $D_i$, but mostly hurt

the overall accuracy on $\mathcal{D}_{tr}$. We therefore define the principal gradient as follows and show how to use it to solve DG model training via gradient descent optimization (where $k$ is a hyperparaemter):

$$\nabla_p = \sum_{z=0}^{k} \frac{\lambda_z ||\nabla_r||_2}{||\boldsymbol{\lambda}[: k]||_2} \boldsymbol{w}_z, \quad \Theta^{t+1} = \Theta^t - \gamma \nabla_p. \tag{12}$$

## 2.2 THEORETICAL ANALYSIS

In Appendix (A.5.1), we prove that $\frac{1}{n}\hat{\boldsymbol{S}}^T\hat{\boldsymbol{S}}$ in Eq. (6) provides us with the mean of all training domains' domain-specific Fisher information matrix (FIM). Since FIM is the negative of Hessian under mild conditions, `PGrad` essentially performs spectrum analysis on the approximated Hessian matrix. Moreover, in Appendix (A.5.2), we show that `PGrad` improves the training efficiency of neural networks by recovering a subspace from the original over-parameterized space $\boldsymbol{\Theta}$. This subspace is built from the top eigenvectors of the approximated Hessian. We visualize the evolution of the eigenvalue distributions in Figure 8.

Our theoretical analysis connects to the machine learning literature that performs spectrum analysis of Hessian matrix (Gur-Ari et al., 2018) and connects its top subspace spanned by the eigenvectors to training efficiency and generalization in neural networks (Wei & Schwab, 2019; Ghorbani et al., 2019; Pascanu et al., 2014). For example, (Hochreiter & Schmidhuber, 1997) shows that small eigenvalues in the Hessian spectrum are indicators of flat directions. Another work (Sagun et al., 2017) empirically demonstrated that the spectrum of the Hessian contains both a bulk component with many small eigenvalues and a few top components of much more significant positive eigenvalues. Later, (Gur-Ari et al., 2018) pointed out that the gradient of neural networks quickly converges to the top subspace of the Hessian.

## 2.3 VARIATIONS OF `PGRAD`

There exist many ways to construct a sampled trajectory, creating multiple variations of `PGrad`.

- `PGrad-F` : The vanilla trajectory sampling method will sample a trajectory of length $n + 1$ by sequentially visiting each $D_i$ in a fixed order. See appendix for the results of the rigid variation.
- `PGrad` : We can randomly shuffle the domain order in the training, and then perform to sample a trajectory. This random order based strategy is used as the default version of `PGrad`.
- `PGrad-B` : We can split each training batch into $B$ smaller batches and construct a long sampled trajectory that is with length $n * B + 1$.
- `PGrad-BMix` : Our method is model and data agnostic. Therefore it is complementary and can combine with many other DG strategies. As one example, we combine the random order based `PGrad-B` and MixUp (Zhang et al., 2017) into `PGrad-BMix` in our empirical analysis.

In `PGrad` and `PGrad-F` , the principal gradient's trajectory covers all training domains $\mathcal{D}_{tr}$ exactly once (per domain). There are two possible limitations. (1) If the number of training domains $n$ is tiny, a length-$(n + 1)$ trajectory will not provide enough information to achieve robustness. In the extreme case of $n = 1$, we will only be able to get one axis $\boldsymbol{w}_z$, that goes back to ERM. (2) The current design can only eliminate discrepancies between different domains. Notable intra-domain variations also exist because empirically approximating the expected loss may include noise due to data sampling, plus batch-based optimization may induce additional bias. Based on this intuition, we propose a new design for sampling a trajectory by evenly splitting $\{\mathbf{x}_k^i, \mathbf{y}_k^i\}$ from a training domain $D_i$ into $B$ small batches. This new strategy allows us to obtain $nB$ pseudo training domains. Such a design brings in two benefits: (1) We can sample a longer trajectory $\boldsymbol{S}$, as the length changes from $n$ to $nB$. (2) Our design splits each domain's data into $B$ batches and treats each batch as if they come from different training domains. By learning the principal gradient from these $nB$ pseudo domains, we also address the intra-domain noise issue. We name this new design `PGrad-B` . Appendix (A.1) includes a figure comparing vanilla trajectory sampling with this extended trajectory sampling.

## 2.4 CONNECTING TO RELATED WORKS

We can categorize existing DG methods into four broad groups.

**Invariant element learning.** Learning invariant mechanisms shared across training domains provides a promising path toward DG. Recent literature has equipped various deep learning components

- especially representation modules - with the invariant property to achieve DG (Li et al., 2018d;e; Muandet et al., 2013). The central idea is to minimize the distance or maximize the similarity between representation distributions $P(f(\mathcal{X})|D)$ across training domains so that prediction is based on statistically indistinguishable representations. Adversarial methods (Li et al., 2018d) and moment matching (Peng et al., 2019; Zellinger et al., 2017) are two promising approaches for distributional alignment. A recent line of work explores the connection between invariance and causality. IRM (Arjovsky et al., 2019) learns an invariant linear classifier that is simultaneously optimal for all training domains. Under the linear case and some constraints, the invariance of the classifier induces causality. Ahuja et al. further extend IRM by posing it as finding the Nash equilibrium (Ahuja et al., 2020) and adding information bottleneck constraints to seek theoretical explanations (Ahuja et al., 2021). However, later works (Kamath et al., 2021) show that even when capturing the correct invariances, IRM still tends to learn a suboptimal predictor. Compared to this stream of works, our method places no assumption on either the marginal or joint distribution. Instead, the `PGrad` explores the promising gradient direction and is model and data-agnostic.

**Optimization methods.** One line of optimization-based DG works is those related to the Group Distributionally robust optimization (a.k.a DRO) (Sagawa et al., 2019). Group DRO aims to tackle domain generalization by minimizing the worst-case training loss when considering all training distributions (rather than the average loss). The second set of optimization DG methods is optimization-based meta-learning. Optimization-based meta-learning uses bilevel optimization for DG by achieving properties like global inter-class alignment (Dou et al., 2019) or local intra-class distinguishability (Li et al., 2018b). One recent work (Li et al., 2019) synthesizes virtual training and testing domains to imitate the episodic training for few-shot learning.

**Gradient manipulation.** Gradient directions drive the updates of neural networks throughout training and are vital elements of generalization. In DG, the main goal is to learn a gradient direction that benefits all training domains (plus unseen domains). Gradient surgery (Mansilla et al., 2021) proposes to use the sign function as a signal to measure the element-wise gradient alignment. Similarly, the authors of (Chattopadhyay et al., 2020) presented And-mask, to learn a binary gradient mask to zero out those gradient components that have inconsistent signs across training domains. Sandmask (Shahtalebi et al., 2021) added a `tanh` function into mask generation to measure the gradient consistency. They extend And-mask by promoting gradient agreement.

Fish (Shi et al., 2021) and Fishr (Rame et al., 2021) are two recent DG works motivated by gradient matching. They require the parallel calculation of the domain gradient from every training domain w.r.t a current parameter vector. Fish maximizes the inner product of domain-level gradients; Fishr uses the variance of the gradient covariance as a regularizer to align the per-domains' Hessian matrix. Our method `PGrad` differs gradient matching by learning a robust gradient direction. Besides, our method efficiently approximates the Hessian with training domains' Fisher information matrix. Appendix (A.4) includes a detailed analysis comparing parallel versus sequential domain-level training. Furthermore, we adapt `PGrad` with parallel training, and compare it against `PGrad` with sequential training and ERM to justify our analysis, see visualizations in Figure 6. We then show that gradient alignment is not necessary a sufficient indicator of the generalization ability in Figure 7.

**Others.** Besides the categories above, there exist other recently adopted to conquer domain generalization. Data augmentation (Xu et al., 2021; Zhang et al., 2021; 2017; Volpi et al., 2018), which generates new training samples or representations from training domains to prevent overfitting. Data augmentation can facilitate a target model with desirable properties such as linearity via Mixup (Zhang et al., 2017) or object focus (Xu et al., 2021). Other strategies, like contrastive learning (Kim et al., 2021), representation disentangling (Piratla et al., 2020), and semi-supervised learning (Li et al., 2021), have also been developed for the DG challenge.

## 3 EXPERIMENTS

We conduct empirical experiments to answer the following questions: **Q1.** Does `PGrad` successfully handle both synthetic and real-life distributional shifts? **Q2.** Can `PGrad` handle various architectures (ResNet and DenseNet), data types (scene and satellite images), and tasks (classification and regression)? **Q3.** Compared to existing baselines, does `PGrad` enable smooth decreasing loss curves and generate smoother parameter trajectories? **Q4.** Can `PGrad` act as a practical complementary approach to combine with other DG strategies? [2] **Q5.** How do bottom eigenvectors in the roll-out trajectories affect the model's training dynamic and generalization ability?

---

[2]Note: we leave hyperparameter tuning details and some ablation analysis results in Appendix (A.7 to A.9).

Table 1: A summary on DOMAINBED dataset, metrics, and architectures we used.

| Dataset | # of Images | Domains | # of Classes |
|---|---|---|---|
| PACS (Li et al., 2017) | 9,991 | Artpaint, Cartoon, Sketches, Photo | 7 |
| VLCS (Fang et al., 2013) | 10,729 | PASCAL VOC 2007, LabelMe, Caltech, Sun | 5 |
| OFFICEHOME (Venkateswara et al., 2017) | 15,588 | Art, Clipart, Product, Real-World | 65 |
| TERRAINCOGNITA (Beery et al., 2018) | 24,788 | Location #100, #38, #43, #46 | 10 |
| DOMAINNET (Peng et al., 2019) | 586,575 | Clipart, Infograph, Painting, Quickdraw, Real, Sketch | 345 |

Table 2: Test accuracy (%) on five datasets from the DomainBed benchmark. We group $20\%$ data from each training domain to construct validation set for model selection.

| Categories | Algorithms | VLCS | PACS | OfficeHome | TerraInc | DomainNet | Avg |
|---|---|---|---|---|---|---|---|
| Baseline | ERM | $77.5 \pm 0.4$ | $85.5 \pm 0.2$ | $66.5 \pm 0.3$ | $46.1 \pm 1.8$ | $40.9 \pm 0.1$ | $63.3$ |
| Invariant | IRM | $78.5 \pm 0.5$ | $83.5 \pm 0.8$ | $64.3 \pm 2.2$ | $47.6 \pm 0.8$ | $33.9 \pm 2.8$ | $61.6$ $_{-1.7}$ |
| | MMD | $77.5 \pm 0.9$ | $84.6 \pm 0.5$ | $66.3 \pm 0.1$ | $42.2 \pm 1.6$ | $23.4 \pm 9.5$ | $58.8$ $_{-4.5}$ |
| | DANN | $78.6 \pm 0.4$ | $83.6 \pm 0.4$ | $65.9 \pm 0.6$ | $46.7 \pm 0.5$ | $38.3 \pm 0.1$ | $62.6$ $_{-0.7}$ |
| | CORAL | $78.8 \pm 0.6$ | $\underline{86.2} \pm 0.3$ | $68.7 \pm 0.3$ | $47.6 \pm 1.0$ | $41.5 \pm 0.1$ | $64.5$ $_{+1.2}$ |
| Optimization | GroupDRO | $76.7 \pm 0.6$ | $84.4 \pm 0.8$ | $66.0 \pm 0.7$ | $43.2 \pm 1.1$ | $33.3 \pm 0.2$ | $60.7$ $_{-2.6}$ |
| | MLDG | $77.2 \pm 0.4$ | $84.9 \pm 1.0$ | $66.8 \pm 0.6$ | $47.7 \pm 0.9$ | $41.2 \pm 0.1$ | $63.6$ $_{+0.3}$ |
| Augmentation | MixUp | $77.4 \pm 0.6$ | $84.6 \pm 0.6$ | $68.1 \pm 0.3$ | $47.9 \pm 0.8$ | $39.2 \pm 0.1$ | $63.4$ $_{+0.1}$ |
| | ARM | $77.6 \pm 0.3$ | $85.1 \pm 0.4$ | $64.8 \pm 0.3$ | $45.5 \pm 0.3$ | $35.5 \pm 0.2$ | $61.7$ $_{-1.6}$ |
| Gradient Manipulation | Fish | $77.8 \pm 0.3$ | $85.5 \pm 0.3$ | $68.6 \pm 0.4$ | $45.1 \pm 1.3$ | $\mathbf{42.7} \pm 0.2$ | $63.9$ $_{+0.6}$ |
| | Fishr | $77.8 \pm 0.1$ | $85.5 \pm 0.4$ | $67.8 \pm 0.1$ | $47.4 \pm 1.6$ | $41.7 \pm 0.0$ | $64.0$ $_{+0.7}$ |
| | PGrad | $\mathbf{79.3} \pm 0.3$ $_{+1.8}$ | $85.1 \pm 0.3$ $_{-0.4}$ | $69.3 \pm 0.1$ $_{+2.8}$ | $49.0 \pm 0.3$ $_{+2.9}$ | $41.0 \pm 0.1$ $_{+0.1}$ | $64.7$ $_{+1.4}$ |
| | PGrad-B | $\underline{78.9} \pm 0.3$ $_{+1.4}$ | $\mathbf{87.0} \pm 0.1$ $_{+1.5}$ | $\underline{69.6} \pm 0.1$ $_{+3.1}$ | $\underline{49.4} \pm 0.8$ $_{+3.3}$ | $41.4 \pm 0.1$ $_{+0.5}$ | $\underline{65.3}$ $_{+2.0}$ |
| | PGrad-BMix | $\underline{78.9} \pm 0.2$ $_{+1.4}$ | $\underline{86.2} \pm 0.4$ $_{+0.7}$ | $\mathbf{69.8} \pm 0.1$ $_{+3.3}$ | $\mathbf{50.7} \pm 0.6$ $_{+4.6}$ | $\underline{42.6} \pm 0.2$ $_{+1.7}$ | $\mathbf{65.7}$ $_{+2.4}$ |

## 3.1 DOMAINBED BENCHMARK

**Setup and baselines.** The DomainBed benchmark (Gulrajani & Lopez-Paz, 2021) is a popular suite designed for rigorous comparisons of domain generalization methods. DomainBed datasets focus on distribution shifts induced by synthetic transformations, We conduct extensive experiments on it to compare it with SOTA methods. The testbed of domain generalization implements consistent experimental protocols for various datasets, and we use five datasets from DomainBed (excluding two MNIST-related datasets) in our experiments. See data details in Table 1.

DomainBed offers a diverse set of algorithms for comparison. Following the categories we summarized in Section 2.4, we compare with invariant element learning works: IRM (Arjovsky et al., 2019), MMD (Li et al., 2018d), DANN (Ganin et al., 2016), and CORAL (Sun & Saenko, 2016). Among optimization methods, we use GroupDRO (Sagawa et al., 2019) and MLDG (Li et al., 2018b). The most closely related works are those based on gradient manipulation, and we compare with Fish (Shi et al., 2021) and Fishr (Rame et al., 2021). Of the representation augmentation methods, we pick two popular works: MixUp (Zhang et al., 2017) and ARM (Zhang et al., 2021). DomainNet's additional model parameters in the final classification layer lead to memory constraints on our hardware at the default batch size of 32. Therefore, we use lower batch size $24$. For our method variation PGrad-B , we set $B = 3$ for all datasets except using $B = 2$ for DomainNet. We default to Adam (Kingma & Ba, 2017) as the optimizer to roll-out a trajectory. All experiments use the DomainBed default architecture, where we finetune a pretrained ResNet50 (He et al., 2016).

**Results analysis.** We aggregate results on each dataset by taking the average prediction accuracy on all domains, and the results are summarized in Table 2. The per-domain prediction accuracy on each dataset is available in Appendix (A.7).

We summarize our observations: 1). ERM remains a strong baseline between all methods, and gradient alignment methods provide promising results compared to other categories. 2). PGrad ranks first out of 11 methods based on average accuracy. Concretely, PGrad consistently outperforms ERM on all datasets and gains $1.8\%$ improvement on VLCS, $2.8\%$ on OfficeHome, $2.9\%$ on TerraIncognita, and no improvement on DomainNet. 3) Our variation PGrad-B outperform PGrad on almost all datasets except VLCS (where it is similar to PGrad ). This observation showcases that intra-domain noise suppression can benefit OOD generalization. A longer trajectory enables PGrad to learn more robust principal directions. 4) The combined variation PGrad-BMix outperforms MixUp across all datasets. On average (see last column of Table 2), PGrad-BMix is the best performing strategy. This observation indicates our method can be effectively combined with other DG categories to improve generalization further.

Table 3: Analysis the effect of varying $k$. The experiments are performed on PACS dataset. We highlight **first** and second best results.

| Method | Algorithms | P | A | C | S | Avg |
|---|---|---|---|---|---|---|
| PGrad | $k=0$ | 98.0±0.2 | 87.3±0.2 | 76.8±0.4 | 73.4±1.3 | 83.9 |
| | $k=2$ | 97.8±0.0 | 87.5±0.3 | 78.2±0.8 | 74.0±1.5 | 84.4 |
| | $k=3$ | 97.8±0.0 | 87.8±0.4 | 78.4±0.6 | 77.2±1.1 | 85.3 |
| | $k=4$ | 97.4±0.1 | 87.6±0.3 | 79.1±1.0 | 76.3±1.2 | 85.1 |
| PGrad-B | $k=0$ | 97.5±0.1 | 89.1±0.8 | 80.3±0.6 | 77.5±0.4 | 86.1 |
| | $k=2$ | 97.7±0.2 | 88.5±1.0 | 79.9±1.1 | 79.2±0.7 | 86.4 |
| | $k=4$ | 98.0±0.2 | **89.9**±0.2 | 80.0±0.6 | **80.1**±0.9 | **87.0** |
| | $k=7$ | 97.6±0.3 | 88.2±0.8 | **81.1**±1.3 | 79.0±1.5 | 86.5 |

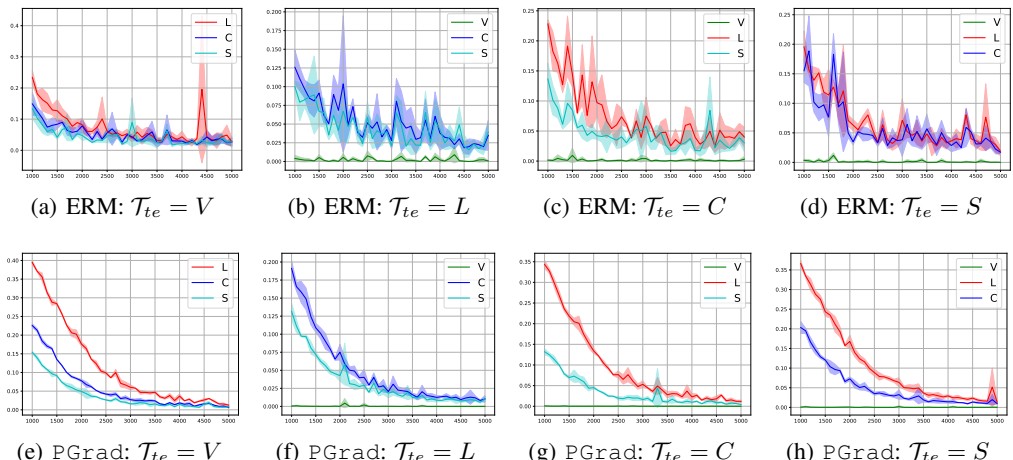

(a) ERM: $\mathcal{T}_{te} = V$    (b) ERM: $\mathcal{T}_{te} = L$    (c) ERM: $\mathcal{T}_{te} = C$    (d) ERM: $\mathcal{T}_{te} = S$

(e) PGrad: $\mathcal{T}_{te} = V$    (f) PGrad: $\mathcal{T}_{te} = L$    (g) PGrad: $\mathcal{T}_{te} = C$    (h) PGrad: $\mathcal{T}_{te} = S$

Figure 2: Visualzing domain-wise training losses on VLCS. Curves are the average over 9 runs, surrounded by $\pm\sigma$ the standard deviation. For comparison, the loss curves start from 1,000 epochs.

**Tuning $k$ for noise suppression.** As we pointed out in Section 2.1, we achieve domain-specific noise suppression by only aggregating coordinate axes $\{\boldsymbol{w}_z\}_{z=0}^{k}$ when learning the principal gradient $\nabla_p$. To investigate the effect of $k$, we run experiments with different values of $k$ for both PGrad and PGrad-B . The analysis results on PACS dataset are collected in Table 3. Note that for default version of PGrad , the maximum number of training domains is $n = 3$, therefore, the length of the PGrad trajectory is upper bounded by 4.

Table 3 shows that the generalization accuracy initially improves and then drops as $k$ increases. If we use $k = n + 1$ (as $k = 4$ for PGrad), domain-specific noise is included and aggregated from principal coordinate $\boldsymbol{W}$ and the performance decreases compared with $k = 3$. The same pattern can also be observed in PGrad-B (note: the length of the trajectory is upper bounded by $nB + 1 = 10$).

**Training loss curve analysis.** Learning and explaining a model update's behavior is an important step toward understanding its generalization ability. To answer **Q3**, we look for insights by visualizing domain-wise training losses as updates proceed. To prevent randomness, we plot average results together with the standard deviation over 9 runs. The results for ERM and PGrad are visualized in Figure 2. Compared to ERM, our method PGrad has smoother decreasing losses as training proceeds. Besides, all training domains benefit from each update in PGrad. On the other hand, ERM's decreasing loss on one domain can come with increases on other domains, especially late in training. We hypothesize this is because domain-specific noise takes a more dominant role as training progresses in ERM. PGrad can effectively suppress domain-specific noise and optimize all training losses in unison without significant conflict. Moreover, the loss variances across training domains are stably minimized, achieving a similar effect as V-REx (Krueger et al., 2021) without an explicit variance penalty. In Appendix (A.2), we visualize four training trajectories trained with PGrad and ERM. ERM trajectories proceed over-optimistically at the beginning and turn sharply in late training. PGrad moves cautiously for every step and consistently towards one direction.

Table 4: A summary on WILDs dataset, metrics, and architectures we used.

| Dataset | Domain Types | Input | Output | Train Domains | Val Domains | Test Domains | Metric | Arch. |
|---|---|---|---|---|---|---|---|---|
| POVERTYMAP | Countries (23), Urban/Rural (2) | Satellite Images | Asset (real valued) | 13 | 5 | 5 | Pearson (r) | ResNet-18 |
| FMoW | Time (16), Regions (5) | Satellite Images | Land Use (62 classes) | 11 | 3 | 2 | Avg Region Acc. | DenseNet-121 |

Table 5: Results on WILDs benchmark. Top two results are highlighted.

| Method | FMoW | | POVERTYMAP | |
|---|---|---|---|---|
| | Val. Accuracy (%) | Test Accuracy (%) | Val. Pearson | Test Pearson |
| IRM | 57.2±0.01 | 50.9±0.32 | 0.81±0.04 | 0.78±0.03 |
| Coral | 56.7±0.06 | 50.5±0.30 | 0.80±0.04 | 0.77±0.05 |
| Fish | 57.3±0.01 | 51.8±0.12 | 0.80±0.01 | 0.80±0.01 |
| PGrad | 57.5±0.15 | 51.9±0.10 | 0.81±6e-3 | 0.80±0.01 |
| PGrad-B | **57.9**±0.08 | **52.1**±0.09 | **0.82**±8e-4 | **0.82**±5e-3 |

## 3.2 WILDs BENCHMARK

WILDs (Koh et al., 2021) is a curated benchmark of 10 datasets covering real-life distribution shifts in the wild such as poverty mapping and land use classification. We apply our method to its two vision applications. Our goal is to investigate the scalability of PGrad under different model architectures and metrics, and more importantly, its performance when facing real-world OOD shifts. We conduct experiments on WILDs to explore both **Q1** and **Q2**.

For each dataset, we use the recommended metrics and model architecture. A summary of the dataset, the metrics, and the model architectures are provided in Table 4. **(I)** The POVERTYMAP dataset is collected for poverty evaluation, where the input $\mathbf{x} \in \mathcal{X}$ is an 8-channel multispectral satellite image, the output $\mathbf{y} \in \mathcal{Y}$ is the real-value asset wealth index. The dataset includes satellite images from 23 different countries and covers both urban and rural areas for each country. We use 13 countries as training domains, pick 5 other countries for model selection, and use the remaining 5 countries for test purpose. We calculate the Pearson correlation $r$ between the predicted index and the groundtruth, and report the average $r$ across 5 test domains and two different areas. **(II)** The objective of the FMoW dataset is to categorize land use based on RGB satellite images spanning 16 years and 5 geographical regions. The training domains contain images from the first 11 years, with the middle 3 years as validation domains and the last 2 years as test domains. We report the average region accuracy on both validation and test domains to evaluate our method under the geographical distributional shift challenge. [3] The training details and the hyperparameters we used can be found in Appendix (A.9). We compare ours with SOTA methods including IRM (Arjovsky et al., 2019), Coral (Sun & Saenko, 2016), and Fish (Shi et al., 2021). We repeat each experiment with three random seeds and report both recommended metrics and their standard deviations on each dataset.

In Table 5, PGrad achieves state-of-the-art results on both datasets, and the variation PGrad-B further improves the performance. On POVERTYMAP, PGrad demonstrates a strong correlation between the predicted wealth index and the ground truth by achieving the highest Pearson coefficient on both validation and testing domains. Its low standard deviation indicates PGrad is stable across different random seeds. PGrad-B achieves better domain generalization by enabling more extended trajectory sampling. Similarly, on the FMoW data, PGrad improves over baseline IRM and Coral, and is on par with Fish. PGrad-B achieves the best accuracy. These experimental results demonstrate that the proposed methods are effective across different model architectures and can successfully handle the real-life distributional shift.

## 4 CONCLUSION

In this paper, we made an effort toward domain generalization by proposing a new training strategy. Our method learns robust gradient directions by analyzing the parameter path created by sequential optimization on training domains. The learned gradient direction, which we call the principal gradient, aggregates and explains the variation of the sampled optimization trajectory. Principal gradients improve domain generalization by reducing the impact of gradient directions specific to individual training domains. Our design also enables convenient gradient normalization and re-calibration for smooth multi-domain training. Empirically, PGrad demonstrates state-of-the-art performance across two DG benchmark suites, covering both synthetic and real-world distribution shifts.

---

[3] We follow the exact same training protocols as in Fish (Shi et al., 2021). On FMoW, we pretrain the model with ERM to a suboptimal starting and then proceed with PGrad .

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

# A  SUPPLEMENTARY FOR `PGrad`: *Learning robust gradients for domain generalization.*

This appendix includes the following contents:

- We include figures describing the different trajectory sampling methods in A.1.

- To understand how `PGrad` changes the optimization path of model's parameters, we show the TSNE projection of parameters during training in A.2. In Figure 4, we visualize the projection for different datasets and test domains.

- The inclusion of the starting point during trajectory sampling is important; we analyze the reason for this effect in A.3.

- `PGrad` differs from related work in that it adopts a sequential training strategy when sampling a trajectory. We include a detailed discussion comparing the parallel training used in gradient matching methods like Fish with our sequential training strategy in A.4. We empirically demonstrate that gradient alignment is not necessarily a sufficient indicator of the generalization ability, see Figure 7.

- To justify the analysis between `PGrad` under sequential training and parallel training, We design experiments comparing `PGrad-B` (Sequential), `PGrad-B` (Parallel), and ERM. The observations are consistent with the intuitive analysis. Using ERM as baseline, `PGrad-B` (Sequential) enlarges clean direction and `PGrad-B` (Parallel) enlarges noise. See Figure 6 in A.4

- We show that our method has two major theoretical contributions. First, instead of being driven by gradient alignment, `PGrad` learns a gradient flow from the tangent space spanned by the eigenvectors of the Hessian matrix - see A.5.1 for the detailed analysis. Second, in A.5.2, we show that `PGrad` has a deep connection to the efficient training of neural networks by projecting the high-dimensional parameter space to some low-dimensional subspace. To further understand the evolution of eigenvalue distributions over time, we visualize their average log values over 1k training step intervals in Figure 8.

- A critical question is how bottom eigenvectors affect the training dynamic and generalization ability. We design experiments showing that those bottom eigenvectors span the normal subspace perpendicular to the tangent space of the loss landscape. Updating the model with any directions lie within the normal subspace will make no changes to the training loss but hurt the generalization ability. See A.6 for details.

- Finally, in A.7, A.8, and A.9, we show the per-domain prediction accuracy of our `PGrad` variants and the experimental details on both benchmarks.

## A.1  ILLUSTRATIVE EXPLANATION OF THE PROPOSED TRAJECTORY SAMPLING METHODS

We compare the three proposed trajectory sampling methods in Figure 3. As vanilla trajectory sampling only considers inter-domain variations, the long trajectory sampling variant splits per-batch to smaller batches to eliminate intra-domain discrepancies from data collection, data sampling, etc.

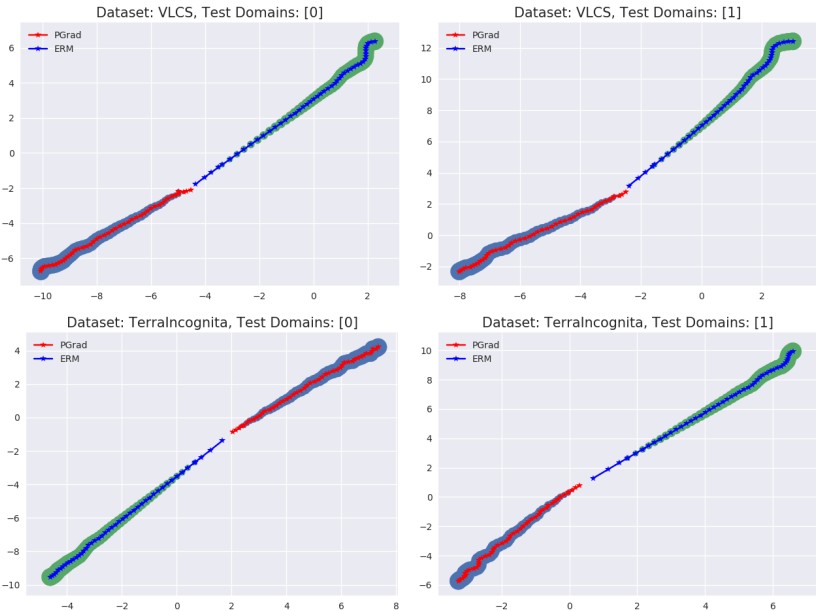

Figure 4: Two-dimensional projection of the parameters' trajectories on different datasets. We use ResNet50 as the backbone and apply TSNE for projection. Both `PGrad` and ERM start from a similar random initialization. Increasing path thickness represents the later training phase. To reinforce the visual effect, we smooth the curve within a window of size 8.

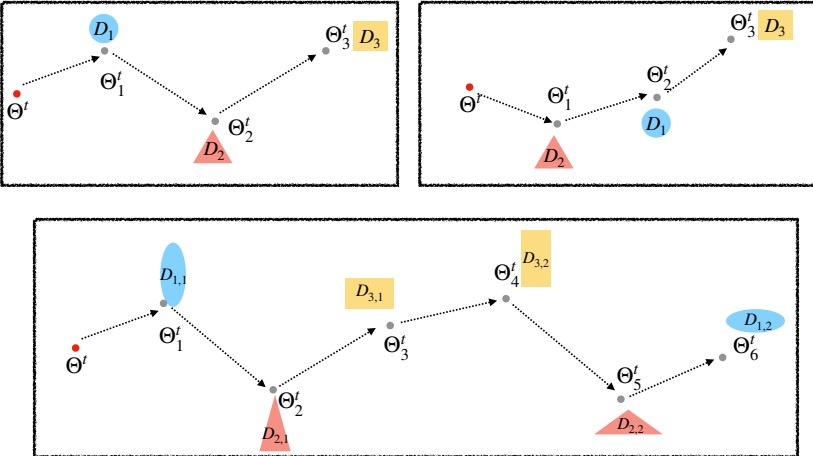

Figure 3: Comparison of three trajectory sampling methods. Assume the number of training domains $n = 3$. The top-left box shows the fixed-order trajectory sampling, which is `PGrad-F`. The top-right box shows the random-order trajectory sampling, the default method `PGrad`. The bottom box represents `PGrad-B`, a version of the long trajectory sampling with $B = 2$.

## A.2 REAL OPTIMIZATION TRAJECTORY VISUALIZATION

We visualize the optimization paths for both our method `PGrad` and the ERM baseline with TSNE projection. Concretely, we save the model's parameters into the memory buffer after every 100 training steps and project them to the $xy$-plane after finishing 5,000 training steps. The trajectories for different datasets and test domains are shown in the following Figure 4.

ERM moves over-confidently with a large step size at the beginning, the turns sharply in late training. Our method `PGrad` 'thinks fast' but 'moves cautiously'. It samples roll-out optimization trajectory

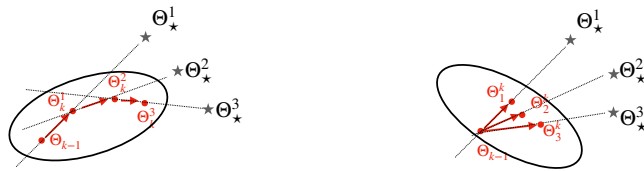

(a): Sequential training will reinforce robust direction    (b): Parallel training will suppress robust direction

Figure 5: Comparison of sequential training and parallel training to learn principal gradient. Stars represent domain-specific optimal minimum, $\Theta^{k-1}$ is the starting point, $\{\Theta_i^k\}_{i=0}^3$ forms a trajectory, the ellipse captures the current principle coordinate chart.

and aggregates the principal directions. The training curves are consistently smooth even late in training.

### A.3 TRAJECTORY SAMPLING

The inclusion of the starting point $\Theta_0^t$ in the trajectory sampling is important, otherwise the learned principal gradient will skip the gradient information from the first training domain from each update. We show the derivation in the following. To simplify the notation, we substitute $\nabla_{\Theta_{i-1}^t} \mathcal{L}_{D_i}$ with $\nabla_{\Theta_i^t}$ and assume we have only two training domains.

Sampled Trajectory With Starting Point $\Theta_0^t$

$$\boldsymbol{S} = \{\Theta_0^t \to (\Theta_0^t - \nabla_{\Theta_1^t}) \to (\Theta_0^t - \nabla_{\Theta_1^t} - \nabla_{\Theta_2^t})\}$$

Trajectory Centering

$$\hat{\boldsymbol{S}} = \{\frac{2\nabla_{\Theta_1^t} + \nabla_{\Theta_2^t}}{3} \to \frac{-\nabla_{\Theta_1^t} + \nabla_{\Theta_2^t}}{3} \to \frac{-\nabla_{\Theta_1^t} - 2\nabla_{\Theta_2^t}}{3}\}$$

Sampled Trajectory W/O Starting Point $\Theta_0^t$

$$\boldsymbol{S} = \{\Theta_1^t \to (\Theta_1^t - \nabla_{\Theta_2^t})\}$$

Trajectory Centering

$$\hat{\boldsymbol{S}} = \{\frac{1}{2}\nabla_{\Theta_2^t} \to -\frac{1}{2}\nabla_{\Theta_2^t}\}$$

After trajectory centering, the gradient information from the first training domain $\nabla_{\Theta_1^t}$ will be skipped if trajectories are sampled without the starting point $\Theta_0^t$. To learn from all training domains, we use the left policy for sampling.

### A.4 COMPARISON BETWEEN THE PARALLEL TRAINING AND SEQUENTIAL TRAINING

In this subsection, we detailed the comparison between our method `PGrad` with the other two gradient-based methods: Fish and Fishr. Both Fish and Fishr are inspired by gradient alignment. They made efforts to align the gradients from different domains and adding the alignment as a penalty to the loss function. In their vanilla implementation, both Fish and Fishr calculate the per-domain gradient w.r.t the current parameter $\Theta_0^t$. We can name the training paradigm as 'parallel training'. As contrast, our method learns a robust gradient direction from the optimization trajectory sampled by sequential training. We explain why parallel training is not a proper choice when learning the principal direction in `PGrad`.

Principal directions learn dominant changing directions. If we apply parallel training in `PGrad`, the centering of the trajectory will suppress the shared pattern and reinforce the domain-specific noises, see Figure 5(b). Instead, the sequential training keeps enlarging shared gradient patterns with each of the multi-step updating. We visualize and compare the two cases using Figure 5. Moreover, the sequential training is more efficient comparing with parallel training.

In Figure 5, we intuitively explain that sequential training will reinforce the learning of a clean direction and parallel training will significantly suppress it. We now design experiments to justify our hypothesis. We adapt `PGrad-B` by learning the principal direction through parallel training. We fix all random mechanisms and compare `PGrad-B` under sequential training, `PGrad-B` under parallel training, and ERM. The test accuracy as functions of the training epoch is visualized in Figure. In the left panel, we use C, P, R as training domains. In the right panel, we use A, P, R as training domains. We attribute the performance drop of the `PGrad-B` (parallel) to the enlarged noise

component and suppressed clean direction. The observations are consistent with above analysis. Another interesting observation is that curves from `PGrad-B` are smoother compared to ERM.

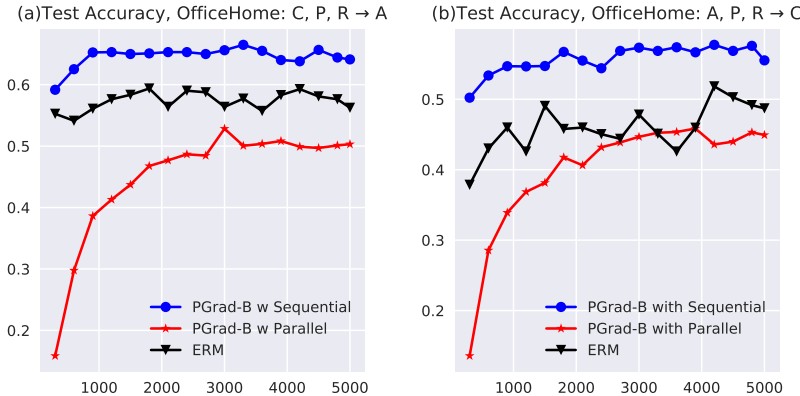

Figure 6: Comparison of `PGrad-B` (sequential) , `PGrad-B` (parallel), and ERM on OfficeHome dataset.

We highlight the novelty of our method by showing that `PGrad` achieves domain generalization without constraining the gradient alignment. Specifically, we define a gradient alignment measurement as the mean of the cosine gradient similarity across all training domain pairs:

$$1/\binom{n}{2} \sum_{i \neq j} \frac{< \nabla(D_i), \nabla(D_j) >}{||\nabla(D_i)||||\nabla(D_j)||} \tag{13}$$

In Figure 7, We visualize the test domain accuracy and training domains gradient alignment as functions of the training epoch. The figure implies the gradient alignment is not a sufficient indicator of the generalization ability. The test accuracy of the `PGrad` is lower bounded by Fish, but Fish aligns training domains' gradient better than `PGrad` . Secondly, the right figure shows the smoothness of the gradient alignment curves has a positive correlation with the test accuracy. Starting from 3,000 epochs, the alignment curve of the `PGrad` becomes smooth, and the model achieves higher prediction accuracy in the test domain. The starting phase of the Fish reveals the same pattern.

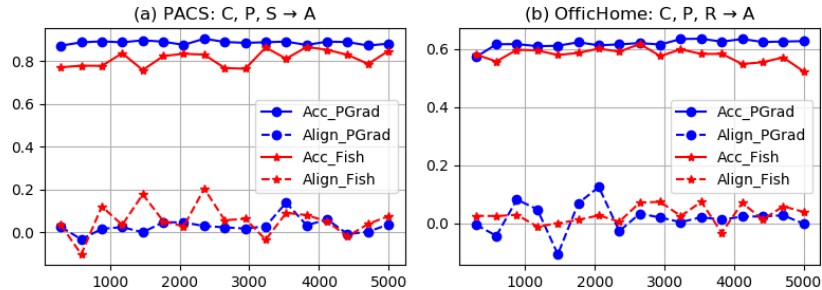

Figure 7: Visualization of test domain accuracy and training domain gradient alignment.

## A.5    THEORETICAL ANALYSIS

The basic motivation of `PGrad` is to learn a gradient flow combines the top eigenvectors of the Hessian which are approximated by the average of the Fisher information matrix calculated sequentially on each training domain.

### A.5.1    CONNECTION TO HESSIAN AND FISHER INFORMATION MATRIX.

Hessian matrix, which is widely used for analyzing the model's training behaviour, is defined as the second order derivative of the function. Similar to NTK (Jacot et al., 2018), we assume the loss

function $\mathcal{L}$ is a functional acting on parameters $\Theta$,

$$\mathcal{L}(\Theta) = \mathbb{E}_{\mathbf{x},\mathbf{y}} \mathcal{L}[\Theta, (\mathbf{x}, \mathbf{y})] \approx \frac{1}{|\mathbf{x} \times \mathbf{y}|} \sum_i \mathcal{L}[\Theta, (\mathbf{x}_i, \mathbf{y}_i)]. \tag{14}$$

We have Taylor expansion around parameter $\Theta$:

$$\mathcal{L}(\Theta') = \mathcal{L}(\Theta) + (\Theta' - \Theta)^T \nabla_\Theta \mathcal{L} + \frac{1}{2}(\Theta' - \Theta)^T \mathcal{H}(\Theta' - \Theta) + \mathcal{O}(||\Theta' - \Theta||^2), \tag{15}$$

where $\mathcal{H}_{i,j} = \dfrac{\partial^2 \mathcal{L}}{\partial \Theta_i \partial \Theta_j}$. The Hessian matrix $\mathcal{H}$ contains local geometric properties of the loss landscape around $\Theta$.

The calculation of the second-order gradient is impractical, especially for modern neural network architectures. Under certain mild regularity conditions and equipped with log-likelihood as loss function (Schervish, 2012), we can approximate $\mathcal{H}$ with the outer product of the gradient. Specifically,

$$\mathcal{I} = \nabla_\Theta \mathcal{L} \otimes \nabla_\Theta \mathcal{L} = -\frac{\partial^2 \mathcal{L}}{\partial \Theta^2} = -\mathcal{H} \tag{16}$$

where $\mathcal{I}$ is also known as Fisher information matrix (FIM).

We explain how our method `PGrad` automatically approximates and aggregates the eigenvalues of the Hessian matrix by following the proposed training procedures. We sample a trajectory as $\boldsymbol{S} = \{\Theta_0, \Theta_1, \cdots, \Theta_n\} \in \mathbb{R}^{(n+1) \times d}$. In the following, we show the trajectory centering operation is equivalent to taking the average of the training domains' Hessian approximations.

**Lemma A.1** *The centered trajectory $\hat{\boldsymbol{S}}$ is the linear transformation of the domain-specific gradient, whose columns can be interpreted as the domain-wise gradient vector starting from the same initialization. The shared initialization is the trajectory center of $\boldsymbol{S}$.*

The first half of the Lemma is easy to show. Any vector within a convex hull can be recovered by the linear combination of its edges. See Appendix A.3 for the derivation on a simple case. It implies that the centered trajectory $\hat{\boldsymbol{S}}$ contains as rich information as training domains' gradient matrix. For the sampled trajectory $\boldsymbol{S}$, the center is calculated as $\boldsymbol{S}_o = \dfrac{\sum_{i=0}^n \Theta_i}{n+1}$. We can re-interpret the sampled trajectory with local coordinate centered with $\boldsymbol{S}_o$. Since the update step is small enough, we have:

$$\hat{\boldsymbol{S}} = [\Theta_0 - \boldsymbol{S}_o, \Theta_1 - \boldsymbol{S}_o, \cdots, \Theta_n - \boldsymbol{S}_o] = [\hat{\nabla}_0, \hat{\nabla}_1, \cdots, \hat{\nabla}_n] \tag{17}$$

The new gradients $\{\hat{\nabla}_i\}_{i=0}^n$ shares pseudo initialization $\boldsymbol{S}_o$.

We then proceed to show the covariance of the centered trajectory gives us the mean of the training domains' FIM.

`PGrad` uses the training domains' average FIM to approximate the real expected FIM.

$$\frac{1}{n}\hat{\boldsymbol{S}}^T \boldsymbol{S} = \frac{1}{n}[\hat{\nabla}_0, \hat{\nabla}_1, \cdots, \hat{\nabla}_n] \otimes [\hat{\nabla}_0, \hat{\nabla}_1, \cdots, \hat{\nabla}_n] \tag{18}$$

$$= \frac{1}{n}\sum_i \hat{\nabla}_i \otimes \hat{\nabla}_i = \frac{1}{n}\sum_i \mathcal{I}_i = -\frac{1}{n}\sum_i \mathcal{H}_i, \tag{19}$$

where $\mathcal{I}_i, \mathcal{H}_i$ are domain-specific FIM and Hessian matrix, respectively. The approximation is a covariance matrix, which is positive semi-definite (SPD) and has the eigenvalue-eigenvector pairs $\{(\lambda_z, \boldsymbol{v}_z)\}_{z=0}^n$ with $\lambda_0 > \lambda_1 > \cdots > \lambda_n$. The eigenvalue $\lambda_z$ is the curvature of the loss in direction of $\boldsymbol{v}_z$ in the neighborhood of $\boldsymbol{S}_o$. The training behaviour of the neural network is determined by the distribution of the eigenvalues. Specifically, first-order optimization methods slow down significantly when $\{\lambda_z\}_{z=0}^n$ are highly spread out (Bottou & Bousquet, 2007; Ghorbani et al., 2019). The property inspires us to zero out insignificant directions and use the directions with the large curvature to derive our gradient direction $\nabla_p$.

As a contrast, Fishr (Rame et al., 2021) uses the current parameter value as the initialization and parallelly approximates the per-domain Hessian matrix with its gradient variance. It defines the Fishr regularization as the square of the Euclidean distance between gradient variance matrices to bring closer domain-wise Hessian. Its goal is to align the second order gradient of the training domains. The high computational cost from parallel training constrains them to **operate only on the last classification layer** in practice.

Comparing with other gradient manipulation works which emphasize *alignment* or *matching*, PGrad uses the average of the FIMs to approximate the underlining Hessian matrix under the DG setup. The operation can also reduce the noise variance by $1/n$, where $n$ is the size of the training domains. The property also provides an explanation of why PGrad-B achieves better generalization ability compared with PGrad. Second, instead of matching eigenvalues of per-domain Hessian, we learn a robust gradient flow which is the combination of the eigenvectors reflecting the dominant changes and zero out the remaining directions, which are empirically proved to be generalization toxic, see Table 3. Our approximation of the Hessian matrix and introduced bijection allows us to **efficiently operate on the high-dimensional parameter space**.

### A.5.2 Connection to the learning behaviour of the neural networks

Modern SOTA neural networks are usually over-parameterized (Allen-Zhu et al., 2019; Zou & Gu, 2019). Improving the training efficiency of high-dimensional neural networks is an active research direction (Li et al., 2018a). Recent works (Gur-Ari et al., 2018; Gressmann et al., 2020) have shown that deep neural networks can be optimized in some subspace of much smaller dimensionality than their native parameter space, we show how PGrad connects to the line of the work.

It was proved by previous work (Li et al., 2018c; Gur-Ari et al., 2018; Gressmann et al., 2020) that the functional induced by the neural network varies most along only some specific directions. We, therefore, focus on recovering the low-dimensional subspace where the loss function $\mathcal{L}$ varies the most on average, and project the native parameter space to the subspace. We formulate the discussion in the following.

Given a direction $\boldsymbol{v}$, the directional derivative of the loss $\mathcal{L}$ at $\Theta$ is defined as:

$$\partial \mathcal{L}_{\boldsymbol{v}}(\Theta) = [\nabla_\Theta \mathcal{L}(\Theta)]^T \boldsymbol{v}, \tag{20}$$

and we can measure the expected scale (or length) of the directional derivative as:

$$\mathbb{E}_\Theta |\partial \mathcal{L}_{\boldsymbol{v}}(\Theta)|^2 = \mathbb{E}_\Theta[\boldsymbol{v}^T \nabla_\Theta \mathcal{L}(\Theta) \otimes \boldsymbol{v}^T \nabla_\Theta \mathcal{L}(\Theta)] = \boldsymbol{v}^T \mathbb{E}_\Theta[\nabla_\Theta \mathcal{L}(\Theta) \otimes \nabla_\Theta \mathcal{L}(\Theta)]\boldsymbol{v}, \tag{21}$$

However, the distribution of the $\Theta$ is unavailable, we turn to utilize the parameters from different training domains for empirical approximation.

$$\mathbb{E}_\Theta |\partial \mathcal{L}_{\boldsymbol{v}}(\Theta)|^2 \approx \boldsymbol{v}^T (\frac{1}{n} \sum_i \hat{\nabla}_n \otimes \hat{\nabla}_n)\boldsymbol{v} = \frac{1}{n}\boldsymbol{v}^T \hat{\boldsymbol{S}}^T \hat{\boldsymbol{S}} \boldsymbol{v}. \tag{22}$$

We demonstrate that learning the principal gradient direction enables us to find a low-rank updating space which is noise resistant by showing the following lemma.

**Lemma A.2** *Suppose we are searching for a $k$ dimensional linear projection $\mathcal{M} \in \mathbb{R}^{k \times d}$ of the original parameter space $\Theta \in \mathbb{R}^d$, such that it keeps largest directional derivatives with respect to the loss $\mathcal{L}$. If the eigenvalue-eigenvector pair of the outer product matrix $\mathbb{E}_\Theta[\nabla_\Theta \mathcal{L}(\Theta) \otimes \nabla_\Theta \mathcal{L}(\Theta)]$ are $\{\lambda_z, \boldsymbol{v}_z\}_{z=0}^n$ with $\lambda_0 > \lambda_1 \cdots > \lambda_n$. We have:*

$$Span\{\mathcal{M}_0, \mathcal{M}_1, \cdots, \mathcal{M}_k\} = Span\{\boldsymbol{v}_0, \boldsymbol{v}_1, \cdots, \boldsymbol{v}_k\}. \tag{23}$$

To understand how eigenvalue changes during training, we visualize the relative contribution of each eigenvalue by normalizing with their summation and then plotting the $log$ of their average over 1k training epochs in Figure 8. In our experiments, we learn $\nabla_p$ by aggregating eigenvectors corresponding to the top-4 eigenvalues. The figure indicates that the contribution from the smallest eigenvalues keeps decreasing as training continues. Our method PGrad can effectively distinguish domain-specific noise and update with shared patterns by suppressing the information from small eigenvalues.

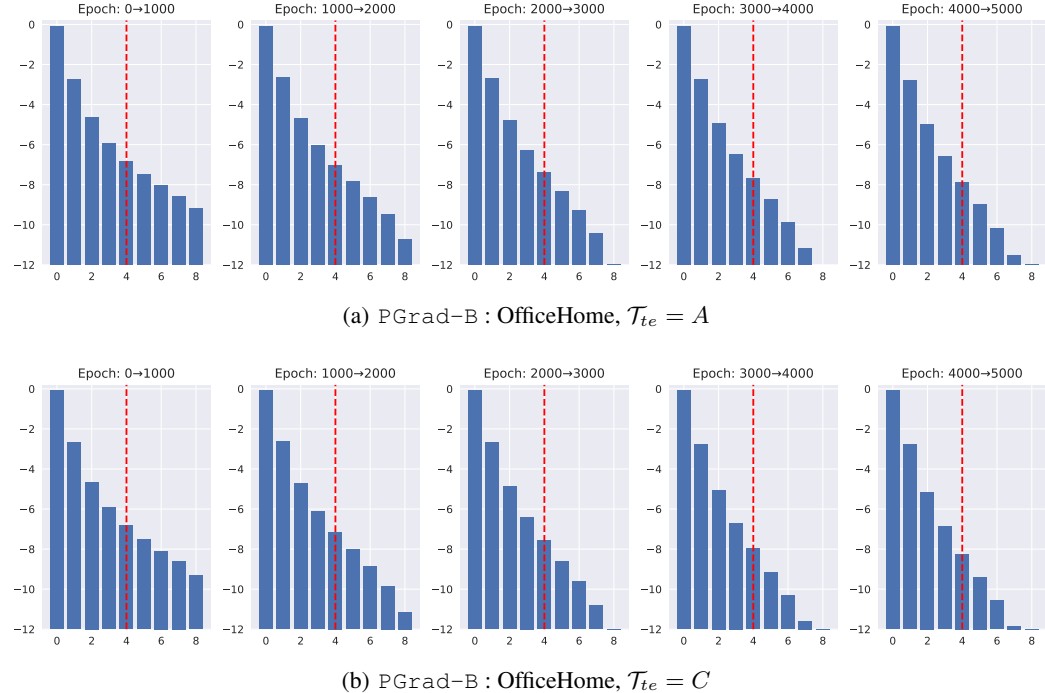

(a) `PGrad-B` : OfficeHome, $\mathcal{T}_{te} = A$

(b) `PGrad-B` : OfficeHome, $\mathcal{T}_{te} = C$

Figure 8: The changing of the eigenvalue distribution with our proposed `PGrad-B` , where $B = 3$. The length of the trajectories is $nB + 1 = 10$. We plot the distribution of the top-9 eigenvalues since the smallest one is out-of-precision. We calculate the contribution of each component by normalization with their sum. To smooth the results, we take the $\log$ value of the average across per-thousand training epochs. The $\nabla_p$ is learned by aggregating the top-4 eigenvectors.

### A.6 WHAT ARE THE TAIL EIGENVECTORS?

To answer the last question we proposed in Section 3, we conduct the analysis to show the effect of the bottom eigenvectors on the training dynamics. Ablation studies in Table 3 indicate that including bottom eigenvalues into principal gradient will hurt the generalization ability. We add new experiments to clarify that the bottom eigenvectors are noise signals of 'special' properties. Concretely, we design three different strategies to update the model with `PGrad` :

• Always from bottom eigenvectors.

• Start from the top eigenvectors and then switch to the bottom vectors in the middle.

• Always from the top eigenvectors.

The training losses keep being constant for case (1) and case (2) after the switching, even when we set the step size to be meaningfully large. The training loss keeps getting decreased for the setup (3). These results imply that those bottom components span the subspace perpendicular to the tangent space of the loss landscape. They do not hurt the training loss but are not helpful for generalization. Similar to the original setup, we calibrate the direction and length for optimization purpose. We show the training loss curves in Figure 9.

### A.7 EXPERIMENTAL SETUP DETAILS FOR DOMAINBED

Following the training protocol described in DomainBed, we run experiments on each domain with a random mechanism to reduce the bias of hyperparameter selection. The DomainBed experiments (Gulrajani & Lopez-Paz, 2021) select the best model from random samples of the hyperparameter search space, and repeat this search process 3 times. Specifically, for each domain, we select 2 random combinations of hyperparameters from a relatively narrow range and repeat the search 3 times to report confidence intervals. For a dataset with $n$ domains, we train $n$ models. $\mathcal{T}_{te}$, held out for testing, and use the rest as training domains $\mathcal{D}_{tr}$. This design leads to a total of $6n$ experiments per dataset for each method.

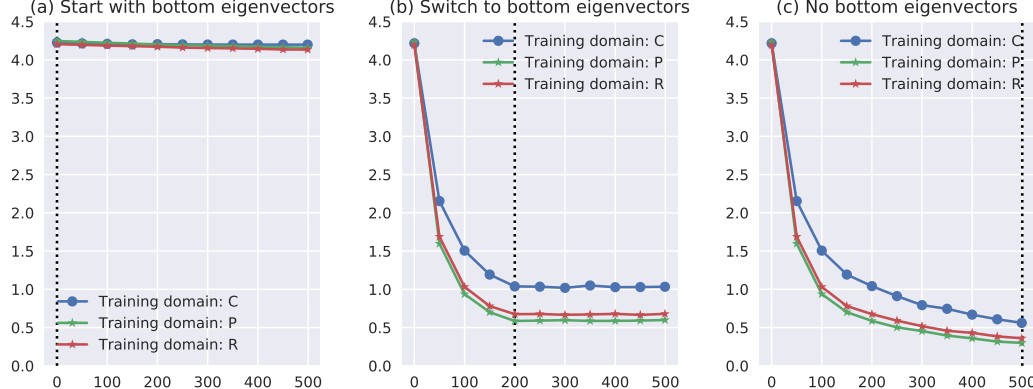

Figure 9: In the left figure, we learn principal gradient always from bottom eigenvectors; the middle figure starts with top eigenvectors and switches to bottom ones after 200 epoch; the right figure always uses top eigenvectors. The vertical line indicates when the intervention happened. The $y$-axis is the training loss, the $x$-axis is the training epoch.

Lacking access to data from the test distribution during training makes it hard to perform model selection for DG compared to other supervised learning tasks. In the main, we follow the popular setup where we sample 20% of the data from each training domain and group them as a validation set. The validation accuracy will be used as an indicator of the optimal model. We show the prediction accuracy of `PGrad` and the extensions on each domain in Tables 7-11.

## A.8 HYPERPARAMETER SEARCH

We following a similar training and evaluation protocol to the DomainBed experiments (Gulrajani & Lopez-Paz, 2021). We list our hyperparameter search space in Table 6.

Table 6: Sample space for hyperparameters.

| Hyper-parameter | Default value | Random distribution |
|---|---|---|
| Inner learning rate | $5e-5$ | $5e-5$ |
| Outer learning rate | 0.1 | 0.1 |
| Batch size | 32 | 32 if not DomainNet else 24 |
| ResNet dropout | 0 | RandomChoice([0, 0.1, 0.5]) |
| Weight decay | 0 | $10^{\text{Uniform}(-6,-4)}$ |
| Training step | 5,000 | 5,000 if not DomainNet else 6,000 |

We design a narrow search space to prevent undersampling and potential performance degeneration from a smaller number of random trials. We use a fixed batch size of 32, which ensures there are adequate samples to provide precise gradient directions when sampling with our long and high-entropy method. For example, there are still 11 samples in each small batch if $B = 3$. We also found the outer learning rate $\gamma = 0.1$ works consistently well on each dataset. DomainNet's additional model parameters in the final classification layer lead to memory constraints on our hardware at the default batch size of 32. Therefore, we use a lower batch sizes 24, and increase the training step to $6,000$.

## A.9 EXPERIMENTAL DETAILS ON WILDS BENCHMARK

A summary of the two vision datasets in the WILDS benchmark is shown in Table 4. For the POVER-TYMAP task, the inputs are 8-channel satellite images, therefore, we tuned the first convolutional layer of the ResNet18 (He et al., 2016) to accommodate a multispectral input. To sample a trajectory of length $\hat{n}+1$ in high dimensional parameter space $\Theta$, we randomly sample $\hat{n}$ training domains and sequentially update the parameter starting from $\Theta^t$ on each of them. We use the Adam optimizer for trajectory construction and set the default learning rate to be $1e-3$ without weight decay. We set the outer learning rate $\gamma = 0.1/\hat{n}$, which adjusts the step size based on the number of sampled training

domains. We train the model with the proposed `PGrad` for 200 epochs and select the optimal model based on the average Pearson coefficient on validation domains.

For the land use classification task FMOW, we follow the exact same training and evaluation protocol applied in Fish (Shi et al., 2021). We find a good starting point by updating the ERM objective with an Adam optimizer for 24,000 iterations with a learning rate of $1e - 4$. After pretraining, we proceed with our proposed `PGrad` and keep tuning the model for 10 epochs with outer learning rate $\gamma = 0.01/\hat{n}$. After training is completed, we report the worst regional accuracy on the test domains. This measurement is designed by WILDS (Koh et al., 2021) to test models' generalization ability when facing both time and regional distribution shift. The numerical results for both datasets are available in Table 5.

Table 7: Per-domain prediction accuracy (%) on VLCS dataset

| Algorithm | C | L | S | V | Avg |
|---|---|---|---|---|---|
| PGrad-F | $98.7 \pm 0.2$ | $64.4 \pm 0.9$ | $73.6 \pm 0.2$ | $76.9 \pm 0.4$ | 78.4 |
| PGrad | $98.3 \pm 0.2$ | $64.4 \pm 0.7$ | $74.4 \pm 0.4$ | $79.9 \pm 0.7$ | 79.3 |
| PGrad-B | $98.7 \pm 0.3$ | $63.9 \pm 1.1$ | $74.6 \pm 0.5$ | $78.5 \pm 0.6$ | 78.9 |
| PGrad-BMix | $99.1 \pm 0.1$ | $63.8 \pm 0.7$ | $73.5 \pm 0.5$ | $79.0 \pm 0.5$ | 78.9 |

Table 8: Per-domain prediction accuracy (%) on PACS dataset

| Algorithm | A | C | P | S | Avg |
|---|---|---|---|---|---|
| PGrad-F | $88.0 \pm 0.4$ | $79.2 \pm 0.3$ | $98.2 \pm 0.2$ | $76.6 \pm 1.5$ | 85.5 |
| PGrad | $87.6 \pm 0.3$ | $79.1 \pm 1.0$ | $97.4 \pm 0.1$ | $76.3 \pm 1.2$ | 85.1 |
| PGrad-B | $89.9 \pm 0.2$ | $80.0 \pm 0.6$ | $98.0 \pm 0.2$ | $80.1 \pm 0.9$ | 87.0 |
| PGrad-BMix | $89.6 \pm 0.3$ | $78.9 \pm 0.6$ | $97.7 \pm 0.3$ | $78.8 \pm 1.0$ | 86.2 |

Table 9: Per-domain prediction accuracy (%) on OfficeHome dataset

| Algorithm | A | C | P | R | Avg |
|---|---|---|---|---|---|
| PGrad-F | $64.4 \pm 0.4$ | $54.7 \pm 0.3$ | $77.0 \pm 0.3$ | $78.6 \pm 0.2$ | 68.7 |
| PGrad | $64.7 \pm 0.6$ | $56.0 \pm 0.7$ | $77.4 \pm 0.2$ | $78.9 \pm 0.3$ | 69.3 |
| PGrad-B | $65.2 \pm 0.4$ | $55.9 \pm 0.8$ | $77.5 \pm 0.5$ | $79.3 \pm 0.3$ | 69.6 |
| PGrad-BMix | $65.8 \pm 0.2$ | $55.4 \pm 0.4$ | $78.0 \pm 0.1$ | $80.0 \pm 0.4$ | 69.8 |

Table 10: Per-domain prediction accuracy (%) on TerraIncognita dataset

| Algorithm | L100 | L38 | L43 | L46 | Avg |
|---|---|---|---|---|---|
| PGrad-F | $52.0 \pm 1.3$ | $42.0 \pm 0.9$ | $57.2 \pm 0.9$ | $43.2 \pm 2.1$ | 48.6 |
| PGrad | $51.2 \pm 0.8$ | $43.4 \pm 0.7$ | $60.0 \pm 0.6$ | $41.3 \pm 0.8$ | 49.0 |
| PGrad-B | $52.7 \pm 2.1$ | $43.5 \pm 0.7$ | $59.5 \pm 0.5$ | $41.9 \pm 0.3$ | 49.4 |
| PGrad-BMix | $61.2 \pm 2.2$ | $45.7 \pm 0.9$ | $58.2 \pm 0.3$ | $37.9 \pm 0.7$ | 50.7 |

Table 11: Per-domain prediction accuracy (%) on DomainNet dataset

| Algorithm | clip | info | paint | quick | real | sketch | Avg |
|---|---|---|---|---|---|---|---|
| PGrad | $57.0 \pm 0.5$ | $18.2 \pm 0.2$ | $48.4 \pm 0.3$ | $13.0 \pm 0.1$ | $60.9 \pm 0.1$ | $48.8 \pm 0.1$ | 41.0 |
| PGrad-B | $57.2 \pm 0.2$ | $18.8 \pm 0.2$ | $48.3 \pm 0.1$ | $13.1 \pm 0.1$ | $61.2 \pm 0.1$ | $49.9 \pm 0.1$ | 41.4 |
| PGrad-BMix | $59.4 \pm 0.1$ | $19.8 \pm 0.1$ | $49.1 \pm 0.4$ | $13.7 \pm 0.1$ | $61.8 \pm 0.2$ | $51.1 \pm 0.1$ | 42.5 |

