# OpenReview forum: "PGrad: Learning Principal Gradients For Domain Generalization"
_ICLR.cc/2023/Conference — ICLR 2023 poster_

### Official Review · Reviewer_jbQd · 2022-10-25

**Confidence:** 4
**Correctness:** 3
**Technical Novelty And Significance:** 3
**Empirical Novelty And Significance:** 3
**Recommendation:** 8

**Clarity, Quality, Novelty And Reproducibility:**

The central assumption of PGrad is that the principal gradient directions, that are common to each batch in a rollout drawn from the different domains available at training time, are those which should provide the best generalization performance to new domains.

Having built on top of DomainBed, the experiments of section 3 should be reproducible (assuming the authors controlled for seeds and other major sources of randomness).  One issue about Table 2 is that it reports the per-domain accuracy average for all domains in all five data sets, possibly masking variation within domains of each task.

Another hinderance to reproducibility is the statement in the header of Table 2 that
> We group 20% data from each training domain to construct validation set for model selection

Gulrajani and Lopez-Paz emphasize that model selection must be very carefully described and carried out for domain generalization.  Do the authors believe this is a sufficiently balanced and detailed way to specify how model selection was carried out?


**Strength And Weaknesses:**

### Strengths
- The paper is written very directly.  The intuition behind the idea of finding principal gradient directions is well explained, and the connection to the estimation of the Fisher Information Matrix is helpful.  Though I will say that Appendix 5.1 is a bit brisk in its explanation.
- The clever trick in equation (7) is compactly explained.
- Section 2 of the paper establishes links to similar ideas in invariant feature learning and gradient regularization.
- Building off of DomainBed makes the method immediately more reproducible and straightforwardly comparable to other domain generalization methods.

### Weaknesses
- In section 2, the authors state that
> When using ERM in DG setting, each step of model updates uses an average gradient and may introduce and preserve domain-specific noise. For instance, if one training domain includes the trapping signals like cows always in pastures and camels always in deserts (as mentioned earlier).

Gulrajani and Lopez-Paz managed to use ERM to achieve great performance on the DomainBed datasets despite being open to this risk.  Given their strong ERM baseline results, which are unable to distinguish domain specific noise from common features, do you think it is a real risk or simply a possible one?

- In section 2.1 just above equation (6), the authors claim “the revised computational complexity is $n$”.
It is unclear what it meant here. The complexity of computing the SVD of the matrix $\hat{\mathbb{S}}^{T}\hat{\mathbb{S}}$ is cubic in the parameter vector size.  Even after introducing the bijection allowing this SVD to be performed on an n x n matrix, the computational complexity is still cubic in $n$.  Could the authors clarify here?
- In section 2.1, equation (9) is justified by scaling the eigenvectors $w_z$ by the ratio of $\frac{\lambda_z}{\mathbf{\lambda}}$.  This is justified under the rubric of normalizing the gradient to reduce training instability, but it is unclear why equation (11) needs to further scale by accounting for the length of $\Delta_r$.  Could the authors provide a footnote or some more justification for this?  Why not simply adjust the learning rate later in training?
-  For equation (12), this exposition can be simplified.  It might help to establish a link between the degree to which each individual $\theta_i$ is optimal (estimated itself via SGD, but not specified IIRC) and the degree to which PGrad can mitigate errors in the directions of any group of $\theta_i$
- Section 3 begins with a series of questions that the authors aim to probe in their experiments.  Question 2 is about whether Pgram will work with different architectures or data types.  I feel like Q2 is almost bound to be answered affirmatively.  How could it not? Any parameterized model will yield gradient estimates, and so the principal gradient can always be computed.  As the authors claim in section 2.4, their model is model and data agnostic.

A question I think would be more impactful, is to see is how well the principal gradient performs in the presence of noisy data that yield high variance gradient estimates?  Could the principal gradient be combined with other variance reduction techniques to yield more stable updates in the presence of such noisy (and variable) gradient estimates?


**Summary Of The Paper:**

This paper addresses the problem of how to train a single model to handle predictions in multiple domains (domain generalization), more specifically how to learn a model that will generalize well across the different domains.  The authors argue that one deficiency of current approaches is that models often fail to lear nt the distinction between shared common features and domain-specific noise, hindering their generalization.  They define the principal gradient (or PGrad), which is derived as the subspace of the gradients which capture the largest directions of variation (akin to PCA), and argue that updating the model by steps in the direction of the principal gradient will allow for greater generalization capabilities.  They show that different variations on PGrad achieve strong results on DomainBed benchmark datasets, and competitive results on WILDS. They also establish with experimental evidence that variation in the training loss is much smoother.



**Summary Of The Review:**

The main technical contributions of this paper are to adapt ideas from principal component analysis to the estimation of gradient directions that are common among disparate domains.  While simple on the surface, the authors show clearly that this is a very clever and elegant solution to domain generalization.

---

> ### Author Response · Authors · 2022-11-16
> **Response to Reviewer jbQd**
>
> We want to thank the reviewer for acknowledging our work. Here are our point-to-point answers.
>
> Q1: Given their strong ERM baseline results, which are unable to distinguish domain specific noise from common features, do you think it is a real risk or simply a possible one?
>
> A1: Thank you for raising this critical question. Our question on whether domain-specific noises are risky is closely related to the question "how bottom eigenvectors affect the training dynamics." In the revised manuscript, we add new experiments to show that bottom eigenvectors span a subspace perpendicular to the tangent space of the loss landscape. Updating the model with any direction from this subspace has no significant effect on the loss values but may hurt the generalization. Please refer to Sec A.6 for more details. Though equipped with strong theoretical guarantees, a large body of existing DG works still can not outperform ERM (as pointed out by the reviewer). Our study showed that PGrad outperforms ERM with promising empirical results across many DG datasets.
>
> Q2: “the revised computational complexity is n”. It is unclear what it meant here.
>
> A2: Thanks for pointing out the error. The correct sentence should have been: the revised computational complexity only depends on $n$. The reviewer’s analysis is correct. Please see the first paragraph on page 4 for a detailed explanation.
>
> Q3: equation (9) is justified by scaling the eigenvectors $w_z$  by the ratio of $\frac{\lambda_z}{||\lambda||_2}$, but it is unclear why equation (11) needs to further scale by accounting for the length of $\nabla_r$.
>
> A3: Great question! Equation 9 indicates that the principal gradient $\nabla_p$ is the convex combination of all oriented and unit-norm eigenvectors. The contribution of each eigenvector is $\frac{\lambda_z}{||\lambda||_2}$. This convex operation generates a unit-norm vector. This property helps us not get ill-conditioned gradients. However, training DNN for thousands of epochs with fixed norm gradients is rigid. We, therefore, have equation (11) to achieve adaptive length-tuning.
>
>
> Q4:  Question 2 is about whether PGrad will work with different architectures or data types. I feel like Q2 is almost bound to be answered affirmatively. How could it not?
>
> A4: Q2 is also a highlight of the PGrad. The data-task-architecture agnostic property makes PGrad more flexible. As a comparison, one line of DG work is prototype-based and is only applicable for classification because it is hard to define a prototype instance for regression.
>
>
> Q5: A question I think would be more impactful, is to see how well the principal gradient performs in the presence of noisy data that yield high variance gradient estimates?
>
> A5: This is a good question, and we agree with the reviewer. PGrad works well when its learned principal gradient points to the shared minimum across all training domains. The perturbation from noisy data at the sample level is loosely related to DG's distribution shift challenge. We leave the adaptation of PGrad for noisy data as our subsequent work.
>
> Q6: One issue about Table 2 is that it reports the per-domain accuracy average for all domains in all five data sets, possibly masking variation within domains of each task.
>
> A6: We show the per-domain accuracy and their standard deviation for all five datasets in Table 7~10. Our method’s domain-wise variations are also more minor compared with other approaches.
>
> Q7: Do the authors believe this is a sufficiently balanced and detailed way to specify how model selection was carried out?
>
> A7: Interesting question! The model selection for DG is still an open problem and is worth exploring. The reference by Gulrajani and Lopez-Paz has a detailed analysis of the topic. Note that we are not proposing a new model-selection method. For a fair comparison, we use the widely-applied way that groups 20% of data from each training domain for validation.

---

> > ### Comment · Reviewer_jbQd · 2022-11-22
> > **Response to authors**
> >
> > >In the revised manuscript, we add new experiments to show that bottom eigenvectors span a subspace perpendicular to the tangent space of the loss landscape. Updating the model with any direction from this subspace has no significant effect on the loss values but may hurt the generalization.
> >
> > Interesting!  However, I don't think that your experiments in Table 3 do enough to establish the claim you make in Appendix A.6.  In A.6, the authors claim that
> > ```
> > Ablation studies in Table 3 indicate that including bottom eigenvalues into principal gradient will hurt the generalization ability
> > ```
> > That's not quite true.  Table 3 reports the effect of increasing the number of principal gradient components on the PACS dataset only; it's hardly enough evidence to empirically support the claim that the bottom eigenvectors span a perpendicular subspace to the tangent space (thought I'm inclined to believe it).  A more convincing experiment would show that as you increase $k$, there is *always* a maximum achieved, followed by some degree of decline, attributable to the effect of including the bottom eigenvector eigenvalue directions in the updates.  Though I will say that Figure 9 presents this information very well!  I would not highlight Table 3 and focus the reader's attention to Figure 9 instead.
> >
> > > This property helps us not get ill-conditioned gradients. However, training DNN for thousands of epochs with fixed norm gradients is rigid. We, therefore, have equation (11) to achieve adaptive length-tuning.
> >
> > Thanks, this makes sense.
> >
> > >one line of DG work is prototype-based and is only applicable for classification because it is hard to define a prototype instance for regression.
> >
> > I didn't realize this in my first reading.  Maybe the authors want to highlight this more emphatically?
> >
> > Overall, I thank you for your thoughtful responses to my review.  My rating is essentially unchanged; it's a solid paper well worthy of acceptance.

---

### Official Review · Reviewer_Fpjy · 2022-10-25

**Confidence:** 4
**Correctness:** 2
**Technical Novelty And Significance:** 3
**Empirical Novelty And Significance:** 2
**Recommendation:** 3

**Clarity, Quality, Novelty And Reproducibility:**

The writing of this paper is clear. The proposed method is novel and should be easy to reproduce.

**Strength And Weaknesses:**

Strength

The technical part of this paper is easy to follow, and the empirical results have shown the potential of the proposed method.

Weaknesses
1. PGrad is motivated by a few intuitions, which are not supported or verified by any theoretical or experimental analysis. For instance, it's unclear why PGrad "forces the learned gradient to filter out domain-specific noise
and follows a direction that maximally benefits all training domains", where the domain-specific noise is not well-defined to begin with.
2. PGrad incorporates multiple techniques, including trajectory sampling and its SVD decomposition, directional calibration, L2 normalization, length calibration, and noise suppression. Without a proper ablation study, it's unclear how each technique contributes to the overall improvement. In particular, it is possible that simple techniques such as sequential optimization on each training domain and gradient normalization are responsible for most of the improvement, in which case the motivation would be even less convincing.
3. The computational overhead of trajectory sampling seems to be significant, but is not discussed in the paper.
4. While Table 2 include 11 methods for comparison, the experiments on the WILDS dataset (Table 5) only include three methods.

**Summary Of The Paper:**

This paper develops a new optimization algorithm named PGrad for domain generalization. The proposed method attempts to filter out domain-specific noise and provide an update direction that maximally benefits all training domains. The experimental results show improved domain generalization performance on multiple datasets.

**Summary Of The Review:**

This paper is easy to follow and has shown some promising results. However, it lacks important theoretical or experimental analysis, and ablation studies.

---

> ### Author Response · Authors · 2022-11-15
> **Response to Reviewer Fpjy**
>
> We thank the reviewer for the interesting questions. First, we would like to clarify that our method is not a stacking technique. Second, we would like to point out that the simple methods mentioned by the reviewer, for instance combining sequential training and gradient normalization,  would not work well for Domain Generalization and are likely to fail, especially on large DG benchmarks.
>
> Q1: PGrad is motivated by a few intuitions, which are not supported or verified by any theoretical or experimental analysis. For instance, it's unclear why PGrad "forces the learned gradient to filter out domain-specific noise and follows a direction that maximally benefits all training domains…
>
> A1: Thank you for the question! We have demonstrated the potential of our method with extensive empirical results, analysis, and visualization in Section 3 and Appendix. To the best of the authors’ knowledge, this paper is the first work to conquer the DG via analyzing the training dynamic through the lens of roll-out trajectories. Theoretically, we connect PGrad to both the spectral analysis of neural networks and the efficient training by subspace learning. We plan to work on the convergence analysis in our next step.
>
> Q2: PGrad incorporates multiple techniques... Without a proper ablation study, it's unclear how each technique contributes to the overall improvement...
>
> A2: Our methods include only two optional operations: noise suppression and trajectory sampling. We used Table 3 and Table 7~10 to demonstrate the results from ablation studies.
>
> For other components, they can not get ablated from the PGrad. This is because:  (1) We learn the principal gradient from the eigenvalues of sampled trajectory. Therefore, trajectory sampling and SVD operation are the core. (2) As illustrated on page 4, all eigenvectors are unoriented. We propose directional calibration to guide the model to climb up or down the loss landscape. So directional calibration is also not optional. (3) Since all eigenvalues are unit-norm vectors, any convex combination is still a unit-norm vector. Therefore, L2 normalization is a desirable property our method intrinsically has. (4) Since the gradient's norm remains constant, we need a weight to adjust the learning rate in training. Therefore, the length calibration component is also not optional.
>
> Q3: The computational overhead of trajectory sampling...
>
> A3:  We measure the model's time efficiency with each update's required time (seconds). We compare the time efficiency of PGrad with one approach from each category of baselines.
>
>               Method      	  P                A         		 C                 S
>               PGrad     	1.20              1.24      	        1.25               1.23
>               PGrad-B   	1.62              1.67                  1.63               1.70
>               Fish    	1.63              1.66                  1.61               1.58
>               DANN      	1.18              1.23                  1.10               1.17
>               MLDG      	1.03 	          1.02                  1.00               1.01
>
> The above table shows that both PGrad and PGrad-B did not introduce significant computational bottlenecks. We will release our code after acceptance.
>
>
> Q4: While Table 2 includes 11 methods for comparison, the experiments on the WILDS dataset (Table 5) only include three methods.
>
> A4: Thanks for the detailed questions. We organize the experiment section with five questions (See the last paragraph on page 6). DomainBed contains five datasets and 22 domains to cover Q3, Q4, and Q5 questions comprehensively.
>
> We use WILDS to answer: Q1. Does PGrad successfully handle both synthetic and real-life distributional shifts? Q2. Can PGrad handle various architectures (ResNet and DenseNet), data types (scene and satellite images), and tasks (classification and regression)? Table 5 includes results for answering these questions.
>
>
>
>
>
> In summary, as noted:
>
> (1) We run ablation studies on optional operations, including noise suppression and trajectory sampling variations;
> (2) Theoretically, we connect PGrad to both the spectral analysis of neural networks and the efficient training by subspace learning;
> (3) Empirically, we provide insights by visualizing a target model's trajectories and the evolving per-domain loss values. We highlight the comparison of sequential versus parallel training. We also analyze the change in the eigenvalue distribution for a more in-depth understanding;
> (4) We add new experiments to analyze how bottom eigenvectors affect the training process and the model's generalization ability.
> We would appreciate it if the reviewer could suggest what type of extra analysis we can run to improve the work.

---

### Official Review · Reviewer_ap7X · 2022-10-28

**Confidence:** 4
**Correctness:** 3
**Technical Novelty And Significance:** 3
**Empirical Novelty And Significance:** 3
**Recommendation:** 8

**Clarity, Quality, Novelty And Reproducibility:**

Quality:
Overall the results of the paper are of high quality. Please see my comments under “Strengths And Weaknesses” for detailed comments.


Novelty:

To the best of my knowledge the approach proposed by the authors (i.e. estimation of robust update direction from principal components of trajectory rollouts in the context of domain generalization) is novel.

The connection between covariance of gradient descent trajectory and the Hessian of the loss (Appendix A.5) has been studied before (see e.g. [1], [2])

I would also like to bring to the authors' attention two prior works which analyze/use PCA of training trajectory for understanding of training dynamics [3] and Bayesian inference in deep networks [4].


[1] Mandt, S., Hoffman, M. D., & Blei, D. M. (2017). Stochastic Gradient Descent as Approximate Bayesian Inference. Journal of Machine Learning Research, 18, 1-35.

[2] Chen, X., Lee, J. D., Tong, X. T., & Zhang, Y. (2020). Statistical inference for model parameters in stochastic gradient descent. The Annals of Statistics, 48(1), 251-273.

[3] Antognini, J., & Sohl-Dickstein, J. (2018). PCA of high dimensional random walks with comparison to neural network training. Advances in Neural Information Processing Systems, 31.

[4] Maddox, W. J., Izmailov, P., Garipov, T., Vetrov, D. P., & Wilson, A. G. (2019). A simple baseline for bayesian uncertainty in deep learning. Advances in Neural Information Processing Systems, 32.

Clarity:

The paper is well-structured and clearly-written. The description of the details of the method and experimental results is clear and easy to understand.

I would recommend proof-reading the paper for typos and grammatical errors. I spotted a few minor presentation issues and typos while reading the paper:
* page 6, typo in 1st paragraph: “momentum matching” -> “moment matching”


* page 6, 4th paragraph. I have a hard time understanding the sentences “Our method PGrad differs gradient matching by designing a robust gradient flow. Besides, our method learns one Hessian under the DG setup.”
* page 6, typo in 5th paragraph “there  exist  other  recent  adopted” -> “there exist other recently adopted” ?


* page 7, Table 2. It would help the reader if the meaning of the subscript numbers in blue, red, and gray was explained in the caption of the table. I understood it from the text of the paper, but it would have been easier if the table was self-explanatory.


* Page 9, typo in the 2nd paragraph: “standard derivations” -> “standard deviations”


* Page 15, 4th paragraph. Perhaps instead of “We visualize the test domain accuracy and training domains gradient alignment in terms of the training epoch.” it would be clearer to say
“We visualize the test domain accuracy and training domains gradient alignment as functions of the training epoch.”

* Page 15, title of section A.5. “Theoretic analysis” -> “Theoretical analysis”?

* Page 16 after equation (16). “We explain how automatically our method PGrad approximates and aggregates the eigenvalues of the Hessian matrix by following the proposed training procedures.” -> “We explain how our method PGrad automatically approximates and aggregates the eigenvalues of the Hessian matrix by following the proposed training procedures.”?


Reproducibility:
I believe that the paper provides sufficient details for other researchers to be able to reproduce the results. I would encourage the authors to release the code for the reproduction of the experiments.


**Strength And Weaknesses:**

Strengths:
* The paper proposes a simple and effective method for domain generalization.
* The evaluation and ablations of the proposed approach are comprehensive.
* The authors thoroughly discuss the connection and differences between the proposed approach and the recent related work. I find the argument about the benefit of the sequential gradient computation approach compared to parallel gradients particularly interesting (Appendix A.4).

Weaknesses:

I did not find any critical weaknesses. Below are some questions that came to my mind when reading the paper. These are not meant to be taken as a criticism of the work done by the authors but rather as open questions and potential directions for future work.

After reading the paper, it is clear that the method delivers improvements on domain generalization benchmarks. However, the mechanisms behind the success of the method and theoretical justification are not completely clear. The authors provide several insightful yet not very concrete and rather intuition-level explanations (such as noise suppression and connection between trajectory of covariance matrix and Hessian of the loss), but some questions remain unanswered:

* Do the bottom eigenvectors actually correspond to “domain-specific” noise? Is there any way to validate this claim directly (theoretically or in experiment)?


* The authors point out that, compared to approaches based on representation alignment or invariance, PGrad does not place any explicit assumptions on the data distributions across domains and the type of distribution shift. On the one hand it is a nice property as the method can be applied to any domain generalization problem in principle, on the other hand it tells us that we do not really know in what scenarios the method is expected to work well and more importantly when it will fail. Clearly, no single method can be successful in all possible domain generalization settings. Figuring out assumptions on the data generating process and the model under which the method is guaranteed to succeed is an important open question.

* What can be said about the method from the optimization perspective? Does PGrad have convergence guarantees?

* How does the wall-clock time of PGrad training compare to that of SGD training? How significant is the computational overhead of PGrad?

* I find the intuitive explanation of the benefit of the sequential training vs parallel training (Figure 5) very interesting. Is it possible to develop this argument in more precise theoretical statements or design experiments highlighting the effect?

I realize that answering most of these questions requires conducting separate investigations, but I would still encourage authors to include a discussion of these issues mentioned above in the revised version of the paper.


**Summary Of The Paper:**

This paper develops a new optimization method for domain generalization. The idea of the proposed PGrad approach is to use a robust gradient direction for parameter update. The robust direction is estimated in 2 steps. First, the rollout of the weights trajectory is collected by sequentially applying training updates in each of the training domains. Then, the principal components of the collected weight space point cloud are estimated and the final update direction is constructed as a combination of the leading principal directions with the downplayed or truncated contribution of the tail directions. The intuition behind this approach is that the low-variance directions are likely to correspond to domain-specific noise and suppression of this noise presumably leads to robust training updates and better generalization on unobserved test domains.

The authors conduct an extensive set of experiments demonstrating the effectiveness of the approach on large scale domain generalization benchmarks: DomainBed and WILDS. The method delivers improvement over recent domain generalization methods including closely related approaches (Fish and Fishr) that are based on cross-domain gradient penalties.


**Summary Of The Review:**

The authors develop a simple and effective method for domain generalization which improves
OOD generalization on large-scale benchmarks.

---

> ### Author Response · Authors · 2022-11-15
> **Response to Reviewer ap7X**
>
> Q1: Do the bottom eigenvectors actually correspond to “domain-specific” noise? Is there any way to validate this claim directly (theoretically or in experiment)?
>
> A1:  Thank you for the insightful question! Motivated by the question, we add new experiments to clarify that the bottom eigenvectors are noise signals of 'special' properties. In summary, we design three different strategies to update the model with PGrad:
>
> (1)Always from bottom eigenvectors.
>
> (2)Start from the top eigenvectors and then switch to the bottom vectors in the middle.
>
> (3)Always from the top eigenvectors.
>
> The training losses keep being constant for case (1) and case (2) after the switching, even when we set the step size to be meaningfully large. The training loss keeps getting decreased for the setup (3). These results imply that those bottom components span the subspace perpendicular to the tangent space of the loss landscape. They do not hurt the training loss but are not helpful for generalization. See ablation studies in Table.3. We have added new figures and analysis to Sec.A.6, highlighted with red font.
>
> In addition, randomness exists in data sampling, batch split, and domain shuffling. It makes finding the training domain that domain-specific noises belong to not practical for batch-based optimization. Suppose the $k$-th eigenvalue originates from the $i$-th domain for the current batch. This relationship will likely not stay true in the next update.
>
> Q2: PGrad does not place any explicit assumptions on the data distributions across domains and the type of distribution shift. On the one hand it is a nice property as the method can be applied to any domain generalization problem in principle, on the other hand it tells us that we do not really know in what scenarios the method is expected to work well and more importantly when it will fail.
>
> A2: Great question! The upside is that we can use PGrad in most domain generalization setups; the downside is that PGrad does not model the data generation process. For instance, PGrad can not handle casual learning. We consider a simplified causal setup: a two-dimensional regression task with two training domains. Assuming a parameterized model has parameters $(w_1, w_2)$ associated with features $(x_1, x_2)$ and data were generated by an unknown data generation graph $x_1 \to y$. With training, the gradients are the same for $x_1$ across the two training domains and are different for $x_2$. A causal model can successfully distinguish the difference and will rely on $x_1$ for prediction. However, PGrad will not capture the difference and still places non-zero weights on both $x_1$ and $x_2$.
>
> Q3: What can be said about the method from the optimization perspective? Does PGrad have convergence guarantees?
>
> A3: Thank you for the recommendation. We plan to work on the convergence analysis as our next work.
>
> Q4: How significant is the computational overhead of PGrad?
>
> A4: We measure the model's time efficiency with each update's required time (seconds). We compare the time efficiency of PGrad with one approach from each category of baselines.
>
>            Method          P         A           C           S
>            PGrad          1.20      1.24       1.25        1.23
>            PGrad-B        1.62      1.67       1.63        1.70
>            Fish    	      1.63      1.66       1.61        1.58
>            DANN           1.18      1.23       1.10        1.17
>            MLDG           1.03      1.02       1.00        1.01
>
>
> The above table shows that both PGrad and PGrad-B did not introduce significant computational bottlenecks. We will release our code after acceptance.
>
> Q5:I find the argument about the benefit of the sequential gradient computation approach compared to parallel gradients particularly interesting…  Is it possible to ... design experiments highlighting the effect?.
>
> A5: We want to thank the reviewer for acknowledging our effort. In Figure 5, we illustrate that sequential training will reinforce learning a clean direction and that parallel training will significantly suppress it.
>
> In our revised version, we add experiments demonstrating the difference between parallel and sequential training. We change PGrad by learning the principal direction with parallel training. The adaptation leads to a clear performance drop regarding training loss, training accuracy, and test accuracy. These observations are consistent with our analysis in Sec A.5: we learn noisy directions with parallel training. In addition, we add Figure 6 comparing PGrad(Sequential), PGrad(Parallel), and ERM in Sec A.5 (the new content is highlighted with red color.)
>
> Thanks to the reviewer for pointing out the presentation issues. We have fixed the typos accordingly.
>
> Finally, we would like to extend thanks again to the reviewer for providing new insights into our work. While we analyze the training dynamics by Hessian analysis and efficient model learning, paper [3] [4] has inspired us with new perspectives.

---

### Decision · Program_Chairs · 2023-01-20

**Decision:**

Accept: poster

**Justification For Why Not Higher Score:**


The approach is quite simple. The ablation experiments are somewhat limited across the many design choices beyond the core components that include trajectory sampling, SVD, and directional calibration. No convergence analysis is provided.

**Justification For Why Not Lower Score:**


A simple and effective approach. It appears empirically effective across several domain generalization datasets albeit gains are a bit small over ERM that nevertheless remains a strong baseline for all DG methods.

**Metareview: Summary, Strengths And Weaknesses:**


The goal of the paper is to learn a single model that generalizes across domains. The authors define a recipe for calculating so-called "principal gradient" based on sampled trajectories of domain updates so as to focus the update on consistent directions. The approach is simple, generally applicable, and appears empirically effective across several domain generalization datasets albeit gains are a bit small over ERM that remains a strong baseline for all DG methods. On the negative side, there's little guidance beyond a helpful (causal) negative example for when the method might or might not work. The ablation experiments are somewhat limited across the many design choices beyond the core components that include trajectory sampling, SVD, and directional calibration. Length calibration, for example, could be performed in many ways. Several PGrad variants are nevertheless tested. No convergence analysis is provided. Figure 6 is helpful in characterizing sequential vs parallel training differences. Figure 9 is helpful for understanding the role of bottom eigenvectors.


**Note From Pc:**

if the above contains the word "oral" or "spotlight" please see: "oral" presentation means -> notable-top-5% and "spotlight" means -> notable-top-25%. As stated in our emails, we are disassociating presentation type from AC recommendations

**Summary Of Ac-Reviewer Meeting:**